# Risk Quadrangle and Robust Optimization Based on Extended $\varphi$-Divergence

## Abstract

The Fundamental Risk Quadrangle (FRQ) is a unified framework linking risk management, statistical estimation, and optimization. Distributionally robust optimization (DRO) based on $\varphi$-divergence minimizes the maximal expected loss, where the maximum is over a $\varphi$-divergence ambiguity set. This paper introduces the *extended* $\varphi$-divergence and the extended $\varphi$-divergence quadrangle, which integrates DRO into the FRQ framework. We derive the primal and dual representations of the quadrangle elements (risk, deviation, regret, error, and statistic). The dual representation provides an interpretation of classification, portfolio optimization, and regression as robust optimization based on the extended $\varphi$-divergence. The primal representation offers tractable formulations of these robust optimizations as convex optimization. We provide illustrative examples showing that many common problems, such as least-squares regression, quantile regression, support vector machines, and CVaR optimization, fall within this framework. Additionally, we conduct a case study to visualize the optimal solution of the inner maximization in robust optimization.

## 1 Introduction

*Distributionally robust optimization* with $\varphi$-divergence ambiguity set (Ben-Tal et al., 2013) minimizes the worst-case expected loss over all probability measures within an ambiguity set defined by $\varphi$-divergence. Ahmadi-Javid (2012) shows that the inner maximization of DRO is a *coherent risk measure* (Artzner et al., 1999), which we refer to as the $\varphi$-divergence risk measure in this study. This result establishes a direct connection between DRO and risk-averse optimization.

*The Fundamental Risk Quadrangle* (Rockafellar and Uryasev, 2013) combines regular risk measures with regular measures of deviation, error and regret in a framework that links optimization with statistics. In particular, regression is shown to be equivalent to deviation minimization. Defined through axioms, FRQ clarifies the relationships between objective functions used in tasks such as classification, portfolio optimization, and regression. Through the dual representation of the quadrangle, these tasks can be interpreted as robust optimization.

It is natural to consider integrating DRO into FRQ by completing the quadrangle for $\varphi$-divergence risk measure. The purpose of the integration is two-fold: FRQ provides insights into DRO by connecting objective functions in various tasks and interprets the tasks as DRO, while the $\varphi$-divergence risk measure inspires the construction of new risk quadrangles. However, $\varphi$-divergence risk measure is coherent, which excludes risk measures widely used in machine learning and finance, such as the mean-standard deviation risk measure . These excluded measures are accommodated within the FRQ framework. By extending the divergence function to the entire real line, we introduce a family of regular risk measures and associated quadrangles, which encompass many important examples.

**Main Contributions.** $(i)$ **Extension of $\varphi$-divergence:** We define the extended $\varphi$-divergence and its associated risk measure, allowing for negative values in the worst-case weight. The extension recovers risk measures used as objective functions in various tasks. A notable example is the mean-standard deviation risk measure associated with the extended $\chi^2$-divergence. $(ii)$ **Completion of Quadrangle:** For the extended $\varphi$-divergence risk measure, we complete the risk quadrangle and derive primal and dual representations for risk, deviation, regret and error. The primal representation facilitates convex optimization formulations. The dual representation provides a robust optimization

(RO) interpretation of the measures associated with extended $\varphi$-divergence, and a DRO interpretation of the measures associated with $\varphi$-divergence. Furthermore, the RO objective functions are upper bounds for their DRO counterparts. A well-known example is that the mean-standard deviation risk measure bounds the $\chi^2$-divergence risk measure. $(iii)$ **Examples and Interpretation:** We provide a range of examples to illustrate that the extended $\varphi$-divergence quadrangle recovers many important quadrangles. The quadrangle elements are used as objective functions in various learning tasks, such as least-squares regression, quantile regression, support vector machines, and CVaR optimization. Through the dual representation, these tasks has a novel RO/DRO interpretation.

**Literature Review.** The connection between DRO and coherent risk measure has been extensively studied in Bayraksan and Love (2015); Dommel and Pichler (2020); Kuhn et al. (2024). Gotoh and Uryasev (2017) studies classification as a risk minimization problem. The dual representation of regular risk measure is studied in Rockafellar and Uryasev (2013). The dual representation of coherent regret is studied in Sun et al. (2020); Rockafellar (2020); Fröhlich and Williamson (2022a;b); Rockafellar (2023). Our study is the first to define the extended $\varphi$-divergence, complete the corresponding risk quadrangle, and unify these developments with robust optimization interpretations for various learning tasks.

## 2 PRELIMINARIES

This section provides the necessary background on the $\varphi$-divergence risk measure and the FRQ. We adopt the following standard notations throughout. Let $(\Omega, \Sigma, P_0)$ be a probability space, where $P_0$ is a reference measure. Let $\overline{\mathbb{R}} = \mathbb{R} \cup \{+\infty\}$ denote the extended set of real numbers. Let $X \in \mathcal{L}^2$ be a real-valued random variable. Expectation and standard deviation of a random variable $X$ with respect to the reference measure is denoted by $\mathbb{E}[X]$ and $\sigma(X)$. The set of all probability measures on $(\Omega, \Sigma)$ is denoted by $\mathcal{P}(\Sigma)$. $||X||_p$ denotes the $\mathcal{L}^p$-norm. $const$ denotes a constant.

### 2.1 $\varphi$-DIVERGENCE AND $\varphi$-DIVERGENCE RISK MEASURE

**Definition 2.1** (Divergence Function). A convex lower semi-continuous function $\varphi : \mathbb{R} \to \overline{\mathbb{R}}$ is a *divergence function* if $(i)\varphi(1) = 0, (ii) \operatorname{dom}(\varphi) = \mathbb{R}, (iii)\varphi(x) = +\infty$ for $x < 0, (iv)1 \in \operatorname{int}(\{x : \varphi(x) < +\infty\}), (v)0 \in \partial\varphi(1)$, where the interior is denoted by $\operatorname{int}$, the subgradient is denoted by $\partial$.

**Definition 2.2** ($\varphi$-Divergence (Csiszár, 1963; Morimoto, 1963)). Consider probability measures $P$ and $P_0$, where $P$ is dominated by $P_0$. For a divergence function $\varphi(x)$, the $\varphi$-*divergence* of $P$ from $P_0$ is defined by $D_\varphi(P||P_0) := \int_\Omega \varphi\left(dP/dP_0\right) dP_0$. Let $Q$ be the Radon–Nikodym derivative $dP/dP_0$. We have $D_\varphi(P||P_0) = \mathbb{E}[\varphi(Q)]$.

**Definition 2.3** ($\varphi$-divergence risk measure (Ahmadi-Javid, 2012; Dommel and Pichler, 2020)). Consider a divergence function $\varphi(x)$. The $\varphi$-*divergence risk measure* is defined by $\mathcal{R}_{\varphi,\beta}(X) = \sup_{P \in \mathcal{P}_{\varphi,\beta}} \mathbb{E}_P[X]$, where $\mathcal{P}_{\varphi,\beta} = \{P \in \mathcal{P}(\Sigma) : D_\varphi(P||P_0) \leq \beta\}$.

For any divergence function of the form $\varphi(x) + k(x-1)$, where $k \in \mathbb{R}$, the resulting $\varphi$-divergence and $\varphi$-divergence risk measure remain unchanged. The condition $(v)$ in Definition 2.1 ensures that the other elements in $\varphi$-divergence quadrangle developed in subsequent sections satisfy the defining axioms. The condition $(iv)$ ensures that $\mathcal{R}_{\varphi,\beta}(X) > \mathbb{E}[X]$.

### 2.2 THE FUNDAMENTAL RISK QUADRANGLE FRAMEWORK

FRQ framework studies closed and convex functionals of random variables. A functional $\rho : \mathcal{L}^2 \to \overline{\mathbb{R}}$ is called *convex* if $\rho\left(\mu X + (1-\mu)Y\right) \leq \mu\rho(X) + (1-\mu)\rho(Y), \ \forall \ X, Y \in \mathcal{L}^2, \ \mu \in [0,1]$, and *closed* if $\left\{X \in \mathcal{L}^2 | \rho(X) \leq c\right\}$ is a closed set $\forall \ c < \infty$. A discussion on functional space can be found in Appendix B.

A risk measure aggregates the overall uncertain cost in $X$ into a number, so that the inequality $\mathcal{R}(X) < C$ models that $X$ is adequately smaller than $C$. For a bet that loses a constant amount, the risk equals the constant. Since a risk measures the undesired outcome, it is more conservative than the expectation. An example is the Markowitz risk $\mathbb{E}[X] + \lambda\sigma(X), \lambda > 0$.

**Definition 2.4** (Regular Risk Measure). A closed convex functional $\mathcal{R} : \mathcal{L}^2 \to \overline{\mathbb{R}}$ is a *regular measure of risk* if it satisfies: (1) $\mathcal{R}(C) = C, \ \forall \ C = const$, (2) $\mathcal{R}(X) > \mathbb{E}[X], \ \forall \ X \neq const$.

A deviation measure quantifies nonconstancy as the uncertainty in $X$ by measuring deviation from the expectation. Intuitively, a bet that loses a fixed amount has risk but zero uncertainty. An example is the (scaled) standard deviation $\lambda\sigma(X)$.

**Definition 2.5** (Regular Deviation Measure). A closed convex functional $\mathcal{D} : \mathcal{L}^2 \to \overline{\mathbb{R}}^+$ is a *regular measure of deviation* if it satisfies: (1) $\mathcal{D}(C) = 0, \ \forall \ C = const$, (2) $\mathcal{D}(X) > 0, \ \forall \ X \neq const$.

Regret quantifies the displeasure associated with the mixture of potential positive, zero and negative outcomes of a random variable. An example is $\mathbb{E}[X] + \lambda||X||_2$.

**Definition 2.6** (Regular Regret Measure). A closed convex functional $\mathcal{V} : \mathcal{L}^2 \to \overline{\mathbb{R}}$ is a *regular measure of regret* if it satisfies: (1) $\mathcal{V}(0) = 0$, (2) $\mathcal{V}(X) > \mathbb{E}[X], \ \forall \ X \neq const$.

An error quantifies nonzeroness. In regression, it measures how wrong an estimate is compared to the true value. It is therefore nonnegative. When the estimate is precise, the error is zero. An example is the scaled $\mathcal{L}^2$ norm $\lambda||X||_2$ used in least squares regression.

**Definition 2.7** (Regular Error Measure). A closed convex functional $\mathcal{E} : \mathcal{L}^2 \to \overline{\mathbb{R}}^+$ is a *regular measure of error* if it satisfies the following axioms: (1) $\mathcal{E}(0) = 0,$, (2) $\mathcal{E}(X) > 0, \ \forall \ X \neq const$.

The measures defined above are intrinsically connected. There is a one-to-one correspondence between risk and deviation, and between regret and error. Risk and deviation can be derived from one-dimensional minimization problems involving regret and error, respectively. When reviewing the axioms, it is helpful to use the provided examples as a guide: risk $\mathbb{E}[X] + \lambda\sigma(X)$, deviation $\lambda\sigma(X)$, regret $\mathbb{E}[X] + \lambda||X||_2$, and error $\lambda||X||_2$.

**Definition 2.8** (Regular Risk Quadrangle). A quartet $(\mathcal{R}, \mathcal{D}, \mathcal{V}, \mathcal{E})$ of regular measures of risk, deviation, regret, and error satisfying the following relationships is called a *regular risk quadrangle*:

(Q1) **error projection:** $\mathcal{D}(X) = \inf_{C}\big\{ \mathcal{E}(X - C) \big\}$;

(Q2) **certainty equivalence:** $\mathcal{R}(X) = \inf_{C}\big\{ C + \mathcal{V}(X - C) \big\}$;

(Q3) **centerness:** $\mathcal{R}(X) = \mathcal{D}(X) + \mathbb{E}[X], \quad \mathcal{V}(X) = \mathcal{E}(X) + \mathbb{E}[X]$.

Moreover, the quartet $(\mathcal{R}, \mathcal{D}, \mathcal{V}, \mathcal{E})$ is bound by the statistic $\mathcal{S}(X)$ satisfying $\mathcal{S}(X) = \arg\min_{C\in\mathbb{R}}\big\{ \mathcal{E}(X - C) \big\} = \arg\min_{C\in\mathbb{R}}\big\{ C + \mathcal{V}(X - C) \big\}$.

A regression problem is defined by minimizing the error of the residual. Error minimization is equivalent to deviation minimization, connecting regression problem to deviation and risk measures.

**Definition 2.9** (Regression). Let $Z_f = Y - f(X) - C$, $\bar{Z}_f = Y - f(X)$, where $C \in \mathbb{R}$, $f$ belongs to a class of functions $\mathcal{F}$. A *regression* problem is defined as $\min_{f\in\mathcal{F}, C} \mathcal{E}(Z_f)$.

**Theorem 2.1** (Error Shaping Decomposition of Regression (Theorem 3.2, Rockafellar et al. (2008)). *The solution to regression in Definition 2.9 is characterized by the prescription that*

$$f, C \in \arg\min_{f, C} \mathcal{E}(Z_f) \ \text{if and only if} \ f \in \arg\min_{f} \mathcal{D}(\bar{Z}_f) \ \text{and} \ C \in \mathcal{S}(\bar{Z}_f).$$

**Definition 2.10** (Conjugate Functional, Risk Envelope, Risk Identifier (Rockafellar and Uryasev, 2013)). Let $\rho : \mathcal{L}^2 \to \overline{\mathbb{R}}$ be a closed convex functional. Then a functional $\rho^* : \mathcal{L}^2 \to \overline{\mathbb{R}}$ is said to be *conjugate* to $\rho$ if $\rho(X) = \sup_{Q\in\mathcal{Q}}\{\mathbb{E}[XQ] - \rho^*(Q)\}, \forall \ X \in \mathcal{L}^2$, where $\mathcal{Q} = \text{dom}(\rho^*)$ is called the *risk envelope* associated with $\rho$, and $Q$ furnishing the maximum in the conjugate $\rho^*$ is called a *risk identifier* for $X$.

## 3 EXTENDED $\varphi$-DIVERGENCE QUADRANGLE

### 3.1 DUAL REPRESENTATION OF EXTENDED $\varphi$-DIVERGENCE QUADRANGLE

This section defines the extended divergence function and its associated risk measure, and completes the risk quadrangle in dual representation for the risk measure. Throughout this paper, if extension is

not explicitly mentioned for divergence function, divergence and risk quadrangle, then the referred object is associated with the non-extended version.

We define the extended divergence function by removing the condition $\varphi(x) = +\infty$ for $x < 0$ in Definition 2.1. We frequently work with natural extensions of the divergence function in this study. For example, the divergence function for the $\chi^2$-divergence is $\varphi(x) = x^2$ for $x > 0$; we extend this by using $\varphi(x) = x^2$ for $x \in \mathbb{R}$ to define the extended $\chi^2$-divergence function.

**Definition 3.1** (Extended Divergence Function). A convex lower semi-continuous function $\varphi : \mathbb{R} \to \overline{\mathbb{R}}$ is an extended divergence function if $(i)\varphi(1) = 0, (ii)\operatorname{dom}(\varphi) = \mathbb{R}, (iii)1 \in \operatorname{int}(\{x : \varphi(x) < +\infty\}), (iv)0 \in \partial\varphi(1)$.

Next, we define the extended $\varphi$-divergence risk measure. We will see that the $\varphi$-divergence risk measure (Definition 2.3) is a special case of the extended version.

**Definition 3.2** (Extended $\varphi$-Divergence Risk Measure). The extended $\varphi$-divergence risk measure is defined by $\mathcal{R}_{\varphi,\beta}(X) = \sup_{Q \in \mathcal{Q}^1_{\varphi,\beta}} \mathbb{E}[XQ]$, where $\mathcal{Q}^1_{\varphi,\beta} = \{Q \in \mathcal{L}^2 : \mathbb{E}[Q] = 1, \mathbb{E}[\varphi(Q)] \leq \beta\}$.

We complete the risk quadrangle in dual representation for the extended $\varphi$-divergence risk measure.

**Definition 3.3** (Dual Representation of Extended $\varphi$-Divergence Quadrangle). For an extended $\varphi$-divergence function and $X \in \mathcal{L}^2$, the dual extended $\varphi$-divergence quadrangle is defined by

$$\mathcal{R}_{\varphi,\beta}(X) = \sup_{Q \in \mathcal{Q}^1_{\varphi,\beta}} \mathbb{E}[XQ], \quad (3.1) \qquad \mathcal{D}_{\varphi,\beta}(X) = \sup_{Q \in \mathcal{Q}^1_{\varphi,\beta}} \mathbb{E}[X(Q-1)], \quad (3.3)$$

$$\mathcal{V}_{\varphi,\beta}(X) = \sup_{Q \in \mathcal{Q}_{\varphi,\beta}} \mathbb{E}[XQ], \quad (3.2) \qquad \mathcal{E}_{\varphi,\beta}(X) = \sup_{Q \in \mathcal{Q}_{\varphi,\beta}} \mathbb{E}[X(Q-1)], \quad (3.4)$$

$$\mathcal{S}_{\varphi,\beta}(X) = \operatorname*{arg\,min}_{C \in \mathbb{R}} \sup_{Q \in \mathcal{Q}_{\varphi,\beta}} \mathbb{E}[(X-C)(Q-1)], \quad (3.5)$$

where

$$\mathcal{Q}^1_{\varphi,\beta} = \{Q \in \mathcal{L}^2 : \mathbb{E}[Q] = 1, \mathbb{E}[\varphi(Q)] \leq \beta\}, \quad \mathcal{Q}_{\varphi,\beta} = \{Q \in \mathcal{L}^2 : \mathbb{E}[\varphi(Q)] \leq \beta\} \quad (3.6)$$

are the envelopes associated with $\mathcal{R}_{\varphi,\beta}(X)$ and $\mathcal{V}_{\varphi,\beta}(X)$ respectively.

The next theorem proves that the dual representation above satisfies the axioms in Section 2.2.

**Theorem 3.1** (Extended $\varphi$-Divergence Quadrangle). *Let $\varphi(x)$ be an extended $\varphi$-divergence function, $X \in \mathcal{L}^2$. The quartet $(\mathcal{R}_{\varphi,\beta}, \mathcal{D}_{\varphi,\beta}, \mathcal{V}_{\varphi,\beta}, \mathcal{E}_{\varphi,\beta})$ defined by 3.1–3.4 is a regular risk quadrangle with the statistic 3.5.*

The proof verifying the axioms, based on Ang et al. (2018); Sun et al. (2020), can be found in Appendix C. After the discussion of the $\varphi$-divergence ambiguity set and the risk envelope $\mathcal{Q}$ in Section 3.2, it will be clear that Theorem 3.1 integrates DRO into the FRQ framework. The coherent risk measure in DRO is a special case of the extended $\varphi$-divergence risk measure. New quadrangles can be constructed by plugging extended $\varphi$-divergences into Definition 3.3. The dual representation provides a robust optimization interpretation for many well-known optimization problems (Section 5).

### 3.2 DISCUSSION ON RISK IDENTIFIER $Q$

Consider the (non-extended) $\varphi$-divergence quadrangle. Although there is no requirement in the envelope 3.6 that $Q \geq 0$, the conditions $\varphi(x) = +\infty$ for $x < 0$ and $\mathbb{E}[\varphi(Q)] \leq \beta$ imply that $Q \geq 0$ almost surely. Define indicator function $\mathcal{I}_A(x) = 1$ if $x \in A$ and 0 otherwise. For every $Q \in \mathcal{Q}^1_{\varphi,\beta}$, we can define a probability measure on $(\Omega, \Sigma)$ by $P_Q(A) = \mathbb{E}[\mathcal{I}_A(\omega)Q(\omega)], A \in \Sigma$. Let $Q_0(\omega) = 1$ be the constant random variable. We have $Q_0 \in \mathcal{Q}^1_{\varphi,\beta}$. Define $P_0 = P_{Q_0}$. By definition, $Q$ is the Radon–Nikodym derivative $dP_Q/dP_0$. Then the condition $\mathbb{E}[\varphi(Q)] \leq \beta$ can be equivalently expressed by $D_\varphi(P||P_0) \leq \beta$. The envelope $\mathcal{Q}^1_{\varphi,\beta}$ has a one-to-one correspondence to a set of probability measures $\mathcal{P}_{\varphi,\beta} = \{P \in \mathcal{P}(\Sigma) : D_\varphi(P||P_0) \leq \beta\}$. The dual representations 3.1 and 3.3 can be equivalently written as

$$\mathcal{R}_{\varphi,\beta}(X) = \sup_{P \in \mathcal{P}_{\varphi,\beta}} \mathbb{E}_P[X], \quad \mathcal{D}_{\varphi,\beta}(X) = \sup_{P \in \mathcal{P}_{\varphi,\beta}} \mathbb{E}_P[X] - \mathbb{E}[X].$$

Next, consider the extended $\varphi$-divergence quadrangle. By extending the divergence function, $Q$ can take negative values. Note that the envelope with $Q \geq 0$ is the necessary and sufficient condition for monotonicity of the convex homogeneous functional associated with such envelope (Rockafellar et al., 2006; Rockafellar and Uryasev, 2013). We therefore forgo the interpretation of $Q$ as a Radon-Nikodym derivative, and the monotonicity of the associated risk measure. Instead, $Q$ can be viewed as (potentially negative) weight on samples. In this case, the minimization of 3.1 can still be interpreted as a robust optimization, where the maximum is over a set of weights. Also, the covariance between random variables $X$ and $Q$ is $\mathrm{cov}(X, Q) = \mathbb{E}[(X - \mathbb{E}X)(Q - \mathbb{E}Q)]$. Since $\mathbb{E}Q = 1$ by 3.6, $\mathrm{cov}(X, Q) = \mathbb{E}[X(Q - 1)]$. The deviation 3.3 can be written as $\sup_{Q \in \mathcal{Q}^1_{\varphi,\beta}} \mathrm{cov}(X, Q)$. Thus the worst-case $Q^*$ tracks $X$ as closely as possible.

### 3.3 Primal Representation of Extended $\varphi$-Divergence Quadrangle

This section derives the primal representations of extended $\varphi$-divergence quadrangle in Definition 3.4 from the dual representations in Definition 3.3. The proof is in Appendix D.

**Definition 3.4** (Primal Representation of Extended $\varphi$-Divergence Quadrangle). For an extended divergence function $\varphi(x)$ and $X \in \mathcal{L}^2$, the Primal Extended $\varphi$-Divergence quadrangle is defined by

$$\mathcal{R}_{\varphi,\beta}(X) = \inf_{\substack{C \in \mathbb{R}, \\ t > 0}} t \left\{ C + \beta + \mathbb{E}\left[ \varphi^*\left( \frac{X}{t} - C \right) \right] \right\}, \tag{3.7}$$

$$\mathcal{D}_{\varphi,\beta}(X) = \inf_{\substack{C \in \mathbb{R}, \\ t > 0}} t \left\{ C + \beta + \mathbb{E}\left[ \varphi^*\left( \frac{X}{t} - C \right) - \frac{X}{t} \right] \right\}, \tag{3.8}$$

$$\mathcal{V}_{\varphi,\beta}(X) = \inf_{t > 0} t \left\{ \beta + \mathbb{E}\left[ \varphi^*\left( \frac{X}{t} \right) \right] \right\}, \tag{3.9}$$

$$\mathcal{E}_{\varphi,\beta}(X) = \inf_{t > 0} t \left\{ \beta + \mathbb{E}\left[ \varphi^*\left( \frac{X}{t} \right) - \frac{X}{t} \right] \right\}, \tag{3.10}$$

$$\mathcal{S}_{\varphi,\beta}(X) = \arg\min_{C \in \mathbb{R}} \inf_{t > 0} t \left\{ \frac{C}{t} + \beta + \mathbb{E}\left[ \varphi^*\left( \frac{X - C}{t} \right) \right] \right\}. \tag{3.11}$$

**Theorem 3.2** (Primal Extended $\varphi$-Divergence Quadrangle). *Let $\varphi(x)$ be an extended divergence function, $X \in \mathcal{L}^2$. Elements of the dual extended $\varphi$-divergence quadrangle in Theorem 3.1 can be presented as 3.7–3.11 in Definition 3.4. The optimal $t$ and $C$ in 3.7–3.11 are attainable.*

The quadrangle elements in primal representation facilitates optimization, since the minimax problem of minimizing the worst-case expectation becomes a minimization with additional scalar variable(s). Furthermore, substituting important extended $\varphi$-divergence functions into the definitions, we recover many risk quadrangles with interpretable expressions (Section 4).

### 3.4 Relation between $\varphi$-Divergence Risk Quadrangle and Extended Version

The risk envelope 3.6 corresponding to a divergence function is a subset of the risk envelope corresponding to its extended version with the same radius $\beta$. Thus, the risk, regret, deviation and error in the $\varphi$-divergence quadrangle are bounded from above by those counterparts in the extended version. Therefore, when the quadrangle elements are used as objective function, the RO is a more conservative version of the corresponding DRO. Furthermore, the conditions $\mathbb{E}[\varphi(Q)] \leq \beta$ and $\mathbb{E}[Q] = 1$ imply that for sufficiently small $\beta$, the value of risk identifier $Q$ cannot be negative. In such case, the $\varphi$-divergence quadrangle becomes equivalent to the extended version.

A well-known special case of the bound of risk measure (Theorem 8.2 of Kuhn et al. (2024)) is that the mean-standard deviation risk measure bounds the $\chi^2$-divergence risk measure

$$\sup_{P \in \mathcal{P}_{\varphi,\beta}} \mathbb{E}_P[X] \leq \mathbb{E}[X] + \sqrt{\beta}\sigma(X),$$

where $\varphi(x) = (x-1)^2$ for $x > 0$ and $+\infty$ if $x \leq 0$. The right-hand side is the primal representation of the extended $\chi^2$-divergence risk measure (Example 2 and 6).

# 4 EXAMPLES OF EXTENDED $\varphi$-DIVERGENCE QUADRANGLE

This section presents important examples of extended $\varphi$-divergences and their corresponding $\varphi$-divergence quadrangles. These quadrangles are derived by substituting the convex conjugates of various extended $\varphi$-divergence functions into the primal representation in Definition 3.4. More examples are listed in Appendix E. The derivations are in Appendix F.

## 4.1 EXTENDED $\varphi$-DIVERGENCE QUADRANGLES

We show that two important risk quadrangles are generated by the extended total variation distance (TVD) and extended $\chi^2$-divergence. The new connection enables the interpretation of robust optimization (Section 5).

**Example 1** (Range-based Quadrangle Generated by Extended TVD). Consider the following extended divergence function and its convex conjugate

$$\varphi(x) = |x - 1|, \quad x \in \mathbb{R}, \quad \varphi^*(z) = \begin{cases} z, & z \in [-1, 1] \\ +\infty, & z \in (-\infty, -1) \cup (1, +\infty) \,. \end{cases}$$

The complete quadrangle is as follows

$$\mathcal{R}_{\varphi,\beta}(X) = \frac{\beta}{2}(\operatorname{ess\,sup} X - \operatorname{ess\,inf} X) + \mathbb{E}[X] = \text{ range-buffered risk, scaled}$$

$$\mathcal{V}_{\varphi,\beta}(X) = \beta \operatorname{ess\,sup} |X| + \mathbb{E}[X] = \mathcal{L}^\infty\text{-regret, scaled}$$

$$\mathcal{D}_{\varphi,\beta}(X) = \frac{\beta}{2}(\operatorname{ess\,sup} X - \operatorname{ess\,inf} X) = \text{ radius of the range, scaled}$$

$$\mathcal{E}_{\varphi,\beta}(X) = \beta \operatorname{ess\,sup} |X| = \mathcal{L}^\infty\text{-error, scaled}$$

$$\mathcal{S}_{\varphi,\beta}(X) = \frac{1}{2}(\operatorname{ess\,sup} X + \operatorname{ess\,inf} X) = \text{ center of range, scaled}$$

We recover the range-based quadrangle in Example 4 of Rockafellar and Uryasev (2013). The divergence function is the extended version of TVD in Example 5.

**Example 2** (Mean Quadrangle Generated by Extended Pearson $\chi^2$-Divergence). Consider the following extended divergence function and its convex conjugate

$$\varphi(x) = (x - 1)^2, \quad \varphi^*(z) = \frac{z^2}{4} + z \,.$$

The complete quadrangle is as follows

$$\mathcal{R}_{\varphi,\lambda}(X) = \mathbb{E}[X] + \sqrt{\beta}\sigma(X), \qquad \mathcal{V}_{\varphi,\lambda}(X) = \mathbb{E}[X] + \sqrt{\beta}\,\|X\|_2\,,$$

$$\mathcal{D}_{\varphi,\lambda}(X) = \sqrt{\beta}\sigma(X), \qquad \mathcal{E}_{\varphi,\lambda}(X) = \sqrt{\beta}\,\|X\|_2\,,$$

$$\mathcal{S}_{\varphi,\lambda}(X) = \mathbb{E}[X].$$

We recover the mean quadrangle in Example 1 of Rockafellar and Uryasev (2013). The solution to the regression (error minimization) is not dependent on $\beta$, since it only scales the error function.

The divergence function is the extended version of the $\chi^2$-divergence function in Example 6. The connection between variance penalty and DRO is studied in Lam (2016); Duchi and Namkoong (2019). Theorem 8.2 of Kuhn et al. (2024) shows that the mean-standard deviation risk measure is an upper bound of $\max_{P \in \mathcal{P}_{\varphi,\beta}} \mathbb{E}_P[X]$. This is a special case of the relation between $\varphi$-divergence quadrangle and its extended version discussed in Section 3.4. Example 6 shows that the risk measure being bounded is the second-order superquantile.

## 4.2 $\varphi$-DIVERGENCE QUADRANGLES

The $\varphi$-divergence risk measures presented in this Section are well-known. We complete the risk quadrangle for these $\varphi$-divergence risk measures in the primal representation, which, apart from Example 3, had not been established. Furthermore, for all examples, the quadrangle establishes a novel connection between regression and DRO (Section 5).

**Example 3** (Quantile Quadrangle Generated by Indicator Divergence). Consider the divergence function and its convex conjugate

$$\varphi(x) = \mathbf{1}_{[0,(1-\alpha)^{-1}]}(x), \qquad \varphi^*(z) = \max\{0, (1-\alpha)^{-1}z\}.$$

We obtain the Quantile Quadrangle:

$$\mathcal{R}_{\varphi,\lambda}(X) = \mathrm{CVaR}_\alpha(X), \qquad\qquad \mathcal{V}_{\varphi,\lambda}(X) = \frac{1}{1-\alpha}\mathbb{E}[X_+],$$

$$\mathcal{D}_{\varphi,\lambda}(X) = \mathrm{CVaR}_\alpha(X) - \mathbb{E}[X], \qquad \mathcal{E}_{\varphi,\lambda}(X) = \mathbb{E}\Big[\frac{\alpha}{1-\alpha}X_+ + X_-\Big],$$

$$\mathcal{S}_{\varphi,\lambda}(X) = \mathrm{VaR}_\alpha(X).$$

The derivation of the risk measure is from Ahmadi-Javid (2012); Shapiro (2017). We recover the quantile quadrangle in Example 2 of Rockafellar and Uryasev (2013). Note that the radius $\beta$ of the divergence ball does not appear in the formula in the primal representation. When $\alpha \to 1$, the quadrangle becomes the worst-case-based quadrangle. When $\alpha \to 0$, the risk measure becomes $\mathbb{E}[X]$, which is not risk averse. $\varphi(x)$ in this case violates Definition 2.1.

**Example 4** (EVaR Quadrangle Generated by Kullback-Leibler Divergence). The divergence function and its convex conjugate are

$$\varphi(x) = x\ln(x) - x + 1, \quad \varphi^*(z) = \exp(z) - 1.$$

The complete quadrangle is as follows:

$$\mathcal{R}_{\varphi,\alpha}(X) = \mathrm{EVaR}_\alpha(X) = \inf_{t>0} t\Big\{\beta + \ln\mathbb{E}\Big[e^{\frac{X}{t}}\Big]\Big\}, \; \mathcal{D}_{\varphi,\alpha}(X) = \inf_{t>0} t\Big\{\beta + \ln\mathbb{E}\Big[e^{\frac{X-\mathbb{E}[X]}{t}}\Big]\Big\},$$

$$\mathcal{V}_{\varphi,\alpha}(X) = \inf_{t>0} t\Big\{\beta + \mathbb{E}\Big[e^{\frac{X}{t}} - 1\Big]\Big\}, \qquad\qquad \mathcal{E}_{\varphi,\alpha}(X) = \inf_{t>0} t\Big\{\beta + \mathbb{E}\Big[e^{\frac{X}{t}} - \frac{X}{t} - 1\Big]\Big\},$$

$$\mathcal{S}_{\varphi,\alpha}(X) = t^* \ln\mathbb{E}\Big[e^{\frac{X}{t^*}}\Big],$$

where $t^*$ is a solution of the following equation $t^*\beta + t^* \ln\mathbb{E}\big[e^{\frac{X}{t^*}}\big] - \mathbb{E}\big[Xe^{\frac{X}{t^*}}\big]/\mathbb{E}\big[e^{\frac{X}{t^*}}\big] = 0$. The risk measure in this quadrangle is studied in Ahmadi-Javid (2012).

**Example 5** (Robustified Supremum-Based Quadrangle Generated by Total Variation Distance). Consider the following divergence function and its convex conjugate

$$\varphi(x) = \begin{cases} |x-1|, & x \geq 0 \\ +\infty, & x < 0 \end{cases}, \quad \varphi^*(z) = \begin{cases} -1 + [z+1]_+, & z \leq 1 \\ +\infty, & z > 1 \end{cases}.$$

The complete quadrangle is as follows:

$$\mathcal{R}_{\varphi,\beta}(X) = \frac{\beta}{2}\operatorname{ess\,sup}(X) + (1-\frac{\beta}{2})\mathrm{CVaR}_{\frac{\beta}{2}}(X), \; \mathcal{V}_{\varphi,\beta}(X) = \inf_{\substack{t>0 \\ t\geq\operatorname{ess\,sup}X}} \Big\{t(\beta-1) + \mathbb{E}\big[X+t\big]_+\Big\},$$

$$\mathcal{D}_{\varphi,\beta}(X) = \mathcal{R}_{\varphi,\beta}(X) - \mathbb{E}[X], \qquad\qquad \mathcal{E}_{\varphi,\beta}(X) = \inf_{t>0}\Big\{t(\beta-1) + \mathbb{E}\big[\big[X+t\big]_+ - X\big]\Big\},$$

$$\mathcal{S}_{\varphi,\beta}(X) = \operatorname{ess\,sup}(X) - 2\mathrm{VaR}_{1-\frac{\beta}{2}}(X).$$

The derivation of the risk measure is studied in Example 3.10 of Shapiro (2017).

**Example 6** (Second-order Quantile-based Quadrangle Generated by Pearson $\chi^2$-divergence). The divergence function and its convex conjugate are

$$\varphi(x) = \begin{cases} (x-1)^2, & x \geq 0 \\ +\infty, & x < 0 \end{cases}, \quad \varphi^*(z) = \begin{cases} \frac{z^2}{4} + z, & z \geq -2 \\ -1, & z < -2 \end{cases} = -1 + \Big(\frac{z}{2}+1\Big)^2 I_{z\geq-2},$$

where $I_{\{\cdot\}} = 1$ if the argument in the bracket is true and $0$ otherwise.

The complete quadrangle is

$$\mathcal{R}_{\varphi,\beta}(X) = \sqrt{(\beta+1)\mathbb{E}\left[(X - q_\beta^{(2)}(X))^2 I_{\{X \geq q_\beta^{(2)}(X)\}}\right]} + q_\beta^{(2)}(X) = \text{second-order superquantile,}$$

$$\mathcal{D}_{\varphi,\beta}(X) = \mathcal{R}_{\varphi,\beta}(X) - \mathbb{E}[X] = \text{ second-order superquantile deviation,}$$

$$\mathcal{V}_{\varphi,\beta}(X) = \inf_{t>0} t\beta + \frac{1}{4t}\mathbb{E}\left[\left((X+2t)^2 - 4t^2\right) I_{\{X+2t>0\}}\right],$$

$$\mathcal{E}_{\varphi,\beta}(X) = \inf_{t>0} t\beta + \frac{1}{4t}\mathbb{E}\left[\left((X+2t)^2 - 4t^2\right) I_{\{X+2t>0\}}\right] - \mathbb{E}[X],$$

$$\mathcal{S}_{\varphi,\beta}(X) = q_\alpha^{(2)}(X) = \text{ second-order quantile,}$$

where $\sqrt{1+\beta} = (1-\alpha)^{-1}$, and the statistic $q_\alpha^{(2)}(X)$ is characterized by the equation

$$1 - \alpha = ||(X - q_\alpha^{(2)}(X))_+||_1 (||(X - q_\alpha^{(2)}(X))_+||_2)^{-1}.$$

The risk measure is a special case of the higher-moment coherent risk measure studied in Krokhmal (2007). The risk, deviation and statistic are the same as those of second-order quantile-based quadrangle in Example 12 of Rockafellar and Uryasev (2013).

# 5 ROBUST OPTIMIZATION INTERPRETATION FOR VARIOUS APPLICATIONS

The primal representation of the extended $\varphi$-divergence quadrangle recovers many important quadrangles, whose elements are used in various tasks such as classification, portfolio optimization and regression. As is discussed in Section 3.1 and 3.2, the dual representation of $\varphi$-divergence quadrangle provides the interpretation as DRO. For the extended $\varphi$-divergence quadrangle, the dual representation provides the interpretation as robust optimization which reweights the samples. We formalize the statement and illustrate it with two important examples.

We consider classification 5.1, portfolio optimization 5.4, and regression 5.7. In portfolio optimization, the portfolio loss is $\boldsymbol{w}^\top \boldsymbol{L}$, where $\boldsymbol{w}$ is the portfolio weight, $\boldsymbol{L}$ is the random loss vector. In classification, given attribute $\boldsymbol{X}$, label $Y$ and decision vector $\boldsymbol{w}$, the margin is defined by $L(\boldsymbol{w}, b) = Y(\boldsymbol{w}^\top \boldsymbol{X} - b)$. $\gamma(\boldsymbol{w})$ is the regularization term. In regression, consider a dependent variable (regressant) $Y$, a vector of independent variables (regressors) $\boldsymbol{X} = (X_1, \ldots, X_d)$, a class of function $\mathcal{F}$ and intercept $C \in \mathbb{R}$. The regression residual is defined by $Z_f = Y - f(\boldsymbol{X}) - C$, and the residual without intercept $C$ is defined by $\bar{Z}_f = Y - f(\boldsymbol{X})$.

Each problem has a robust optimization interpretation (5.5, 5.2,5.8) through the dual representations. The equivalence between 5.7 and 5.8 is a result of Theorem 2.1. When $\varphi(x)$ is a divergence function as defined in Definition 2.1, these problems also have a DRO interpretation (5.6, 5.3, 5.9).

| **Classification** | **Robust expected margin maximization** | **DR expected margin minimization** |
|---|---|---|
| $\min_{\boldsymbol{w}} \mathcal{R}_{\varphi,\beta}(-L(\boldsymbol{w}, b))$ $+ \gamma(\boldsymbol{w})$, (5.1) | $\min_{\boldsymbol{w}} \max_{Q \in \mathcal{Q}_{\varphi,\beta}} \mathbb{E}[-QL(\boldsymbol{w}, b)]$ $+ \gamma(\boldsymbol{w})$. (5.2) | $\min_{\boldsymbol{w}} \max_{P \in \mathcal{P}_{\varphi,\beta}} \mathbb{E}_P[-L(\boldsymbol{w}, b)]$ $+ \gamma(\boldsymbol{w})$. (5.3) |
| **Portfolio Optimization** | **Robust loss minimization** | **DRO** |
| $\min_{\mathbf{1}^\top \boldsymbol{w}=1} \mathcal{R}_{\varphi,\beta}(\boldsymbol{w}^\top \boldsymbol{L})$, (5.4) | $\min_{\mathbf{1}^\top \boldsymbol{w}=1} \max_{Q \in \mathcal{Q}_{\varphi,\beta}} \mathbb{E}[Q\boldsymbol{w}^\top \boldsymbol{L}]$. (5.5) | $\min_{\mathbf{1}^\top \boldsymbol{w}=1} \max_{P \in \mathcal{P}_{\varphi,\beta}} \mathbb{E}_P[\boldsymbol{w}^\top \boldsymbol{L}]$ (5.6) |
| **Regression** | **Deviation minimization** | **Deviation minimization** |
| $\min_{f \in \mathcal{F}, C} \mathcal{E}_{\varphi,\beta}(Z_f))$, (5.7) | $\min_f \left\{ \max_{Q \in \mathcal{Q}_{\varphi,\beta}} \mathbb{E}[Q\bar{Z}_f] - \mathbb{E}[\bar{Z}_f] \right\}$ calculate $C = \mathcal{S}(\bar{Z}_f)$ (5.8) | $\min_f \left\{ \max_{P \in \mathcal{P}_{\varphi,\beta}} \mathbb{E}_P[\bar{Z}_f] - \mathbb{E}[\bar{Z}_f] \right\}$ calculate $C = \mathcal{S}(\bar{Z}_f)$ (5.9) |

## 5.1 Examples of Robust Optimization Interpretation

**Example 7** (Mean Quadrangle)**.** The risk and error measure in Mean Quadrangle are objective functions for Large Margin Distribution Machine (Zhang and Zhou, 2014), Markowitz portfolio optimization (Markowitz, 1952), and least squares regression. We have the following robust optimization interpretation.

**Large Margin Distribution Machine**

$$\min_{\boldsymbol{w},b} \mathbb{E}[-L(\boldsymbol{w},b)] + \sqrt{\beta}\sigma(-L(\boldsymbol{w},b)) + \gamma(\boldsymbol{w}),$$

**Robust expected margin maximization**

$$\min_{\boldsymbol{w},b} \max_{Q \in \mathcal{Q}^1_{\varphi,\beta}} \mathbb{E}[-QL(\boldsymbol{w},b)] + \gamma(\boldsymbol{w}).$$

**Markowitz portfolio optimization**

$$\min_{\mathbf{1}^\top \boldsymbol{w}=1} \mathbb{E}[\boldsymbol{w}^\top \boldsymbol{L}] + \sqrt{\beta}\sigma(\boldsymbol{w}^\top \boldsymbol{L}),$$

**Robust expected loss minimization**

$$\min_{\mathbf{1}^\top \boldsymbol{w}=1} \max_{Q \in \mathcal{Q}^1_{\varphi,\beta}} \mathbb{E}[Q(\boldsymbol{w}^\top \boldsymbol{L})] .$$

**Least squares regression**

$$\min_{f \in \mathcal{F}, C \in \mathbb{R}} \sqrt{\beta}||Z_f||_2 ,$$

**Deviation minimization**

$$\min_{f \in \mathcal{F}} \max_{Q \in \mathcal{Q}^1_{\varphi,\beta}} \mathbb{E}[Q\bar{Z}_f] - \mathbb{E}[\bar{Z}_f]$$

$$\text{calculate } C = \mathbb{E}[\bar{Z}_f] .$$

**Example 8** (Quantile Quadrangle)**.** The risk and error measure in Quantile Quadrangle are objective functions of $\nu$-support vector machine (Schölkopf et al., 2000), CVaR optimization (Rockafellar and Uryasev, 2000), and quantile regression (Koenker and Bassett Jr, 1978). Let $\nu = 1 - \alpha$. The equivalence of $\nu$-SVM and CVaR optimization is studied by Gotoh and Takeda (2004); Takeda and Sugiyama (2008). We have the following DRO interpretation.

**CVaR portfolio optimization**

$$\min_{\mathbf{1}^\top \boldsymbol{w}=1} \text{CVaR}_\alpha(X(\boldsymbol{w})) , \qquad (5.10)$$

**DR loss minimization**

$$\min_{\mathbf{1}^\top \boldsymbol{w}=1} \max_{P \in \mathcal{P}_{\varphi,\beta}} \mathbb{E}_P[X(\boldsymbol{w})] , \qquad (5.11)$$

**$\nu$-SVM**

$$\min_{\boldsymbol{w},b} \text{CVaR}_\alpha(-L(\boldsymbol{w},b)) + \gamma(\boldsymbol{w}), \qquad (5.12)$$

**DR expected margin maximization**

$$\min_{\boldsymbol{w}} \max_{P \in \mathcal{P}_{\varphi,\beta}} \mathbb{E}_P[-L(\boldsymbol{w},b)] + \gamma(\boldsymbol{w}) . \qquad (5.13)$$

**Quantile regression**

$$\min_{f \in \mathcal{F}, C \in \mathbb{R}} \mathcal{E}_\alpha(Z_f) , \qquad (5.14)$$

**Deviation minimization**

$$\min_{f \in \mathcal{F}} \max_{P \in \mathcal{P}_{\varphi,\beta}} \mathbb{E}_P[\bar{Z}_f] - \mathbb{E}[\bar{Z}_f] \qquad (5.15)$$

$$\text{calculate } C \in \text{VaR}_\alpha[\bar{Z}_f] , \qquad (5.16)$$

where $X_+ = \max\{0, X\}$, $X_- = \max\{0, -X\}$, $\mathcal{E}(X) = \left[\frac{\alpha}{1-\alpha}X_+ + X_-\right]$ is the normalized Koenker-Bassett error, VaR is Value-at-Risk.

## 6 Statistic and Risk Identifier

This section derives an expression for the risk identifier. Proposition 6.1 allows us to directly calculate the risk identifier (worst-case weight) given the solution to the problem in primal representation. It will be used for calculation in Section 8. The proofs are in Appendix H.

**Proposition 6.1.** *Denote by $C^*$ and $t^*$ the optimal $C$ and $t$ in the primal representation 3.7 of extended $\varphi$-divergence risk measure. The risk identifier of risk measure $\mathcal{R}_{\varphi,\beta}(X)$ can be expressed as $Q^*(\omega) \in \partial\varphi^* \left(X(\omega)/t^* - C^*\right)$. Denote by $C^*$ the optimal $C$ in the primal representation 3.10 of extended $\varphi$-divergence error measure. The risk identifier of extended $\varphi$-divergence error measure $\mathcal{E}_{\varphi,\beta}(X)$ can be expressed as $Q^*(\omega) \in \partial\varphi^* \left(X(\omega)/t^*\right)$.*

## 7 Recovering $\varphi$-divergence from Quadrangle Elements

This study starts with developing new risk measures given a $\varphi$-divergence function. There exists a duality between divergence and risk that allows us to recover the $\varphi$-divergence from the elements of

the corresponding $\varphi$-divergence quadrangle. The proof based on Föllmer and Knispel (2011) is in Appendix I.

**Proposition 7.1.** *Let $\varphi(x)$ be a divergence function. $\varphi$-divergence can be recovered from the elements in $\varphi$-divergence quadrangle by*

$$D_\varphi(P\|P_0) = \sup_{\substack{X\in\mathcal{L}^2 \\ \beta>0}} \{\mathbb{E}[XQ] - \mathcal{R}_{\varphi,\beta}(X) - \beta\} = \sup_{\substack{X\in\mathcal{L}^2 \\ \beta>0}} \{\mathbb{E}[X(Q-1)] - \mathcal{D}_{\varphi,\beta}(X) - \beta\}$$

$$= \sup_{\substack{X\in\mathcal{L}^2 \\ \beta>0,C}} \{\mathbb{E}[XQ] - \mathcal{V}_{\varphi,\beta}(X-C) + C - \beta\} = \sup_{\substack{X\in\mathcal{L}^2 \\ \beta>0,C}} \{\mathbb{E}[X(Q-1)] - \mathcal{E}_{\varphi,\beta}(X-C) - \beta\}.$$

# 8 CASE STUDY: RISK IDENTIFIER VISUALIZATION

This section contains three case studies visualizing the risk envelope (worst-case weight) in classification 5.2, portfolio optimization 5.5, and regression 5.8. We focus on the mean quadrangle (Example 2) in this case study. The details of the experiments are specified in Appendix K.

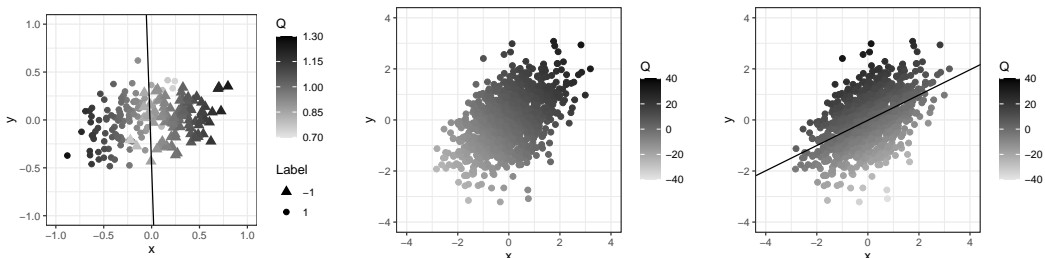

(a) Risk envelope in Large Margin Distribution Machine.

(b) Risk envelope in Markowitz portfolio optimization.

(c) Risk envelope in least squares regression.

Figure 1: Darker points correspond to higher values of $Q^*(w)$ in all figures. In **(a)**, the circles represent samples with label 1, while the diamonds represent samples with label $-1$. The optimal decision line is $y = -28.826x - 0.486$. In **(b)**, optimal portfolio weights $= (0.4999038, 0.5000962)$. In **(c)**, the straight line is the least squares regression line $y = 0.495x - 0.0127$.

From the minimax formulation in the dual representation 3.3, we observe that a larger incurred loss corresponds to a larger weight being assigned to the data point. This observation is confirmed by the figures. In classification 1a, misclassified points with a large margin are assigned larger weights. In portfolio optimization 1b, points in the upper-right corner, corresponding to large portfolio losses, are assigned larger weights. In regression 1c, points further above the regression line are assigned larger weights.

# 9 CONCLUSION

We introduce the extended $\varphi$-divergence risk measure and complete its associated risk quadrangle. The inner maximization problem of DRO is integrated as a special case of the risk measure. The extended $\varphi$-divergence quadrangle encompasses many important quadrangles, whose elements are used as objective functions in well-known learning tasks in classification, portfolio optimization, and regression. The FRQ framework connects the elements axiomatically and provides a RO/DRO interpretation to the tasks. A case study is conducted to visualize the worst-case weight.

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

# A    QUADRANGLE THEOREM

Rockafellar and Uryasev (2013) introduced measures of uncertainty that are built upon the concept of regularity, which is closely linked to convexity and closedness.

Uncertainty can be modeled via random variables and by studying and estimating the statistical properties of these random variables, we can estimate the risk in one form or the other. When the aim is to estimate the risk, it is convenient to think of the the random variable as 'loss' or 'cost'. There are various ways in which risk can be quantified and expressed. One such framework developed by Rockafellar and Uryasev (2013) is called the *Risk Quadrangle*, which is shown in Figure 2.

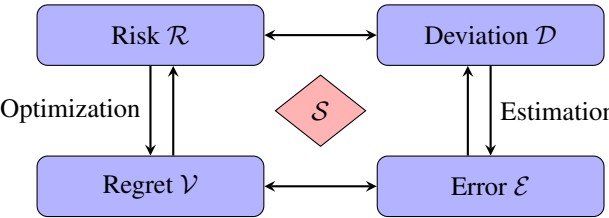

Figure 2: Risk Quadrangle Flowchart

The quadrangle begins from the upper left corner which depicts the *measure of risk* denoted by $\mathcal{R}$. It aggregates the uncertainty in losses into a numerical value $\mathcal{R}(X)$ by the inequality $\mathcal{R}(X) \leq C$ where $C$ is the tolerance level for the risk. The next term is in the upper-right corner called the *measure of deviation* denoted by $\mathcal{D}$ and it quantifies the nonconstancy of the random variable. The lower-left corner depicts *measure of regret* denoted by $\mathcal{V}$. It stands for the net displeasure perceived in the potential mix of outcomes of a random variable "loss" which can be bad ($> 0$) or acceptable/good ($\geq 0$). The last measure is the *measure or error* which sits as the right-bottom of the quadrangle denoted by $\mathcal{E}$. Error quantifies the non-zeroness in the random variable.

**Theorem A.1** (Quadrangle Theorem, Rockafellar and Uryasev (2013)). *The theorem states the following:*

(a) *The centerness relations $\mathcal{D}(X) = \mathcal{R}(X) - \mathbb{E}[X]$ and $\mathcal{R}(X) = \mathbb{E}[X] + \mathcal{D}(X)$ give a one-to-one correspondence between regular measures of risk $\mathcal{R}$ and regular measures of deviation $\mathcal{D}$. In this correspondence, $\mathcal{R}$ is positively homogeneous if and only if $\mathcal{D}$ is positively homogeneous. On the other hand, $\mathcal{R}$ is monotonic if and only if $\mathcal{D}(X) \leq \sup X - \mathbb{E}[X]$ for all $X$.*

(b) *The relations $\mathcal{E}(X) = \mathcal{V}(X) - \mathbb{E}[X]$ and $\mathcal{V}(X) = \mathbb{E}[X] + \mathcal{E}(X)$ give a one-to-one correspondence between regular measures of regret $V$ and regular measures of error $\mathcal{E}$. In this correspondence, $\mathcal{V}$ is positively homogeneous if and only if $\mathcal{E}$ is positively homogeneous. On the other hand, $\mathcal{V}$ is monotonic if and only if $\mathcal{E}(X) \leq |\mathbb{E}[X]|$ for $X \leq 0$.*

(c) *For any regular measure of regret $\mathcal{V}$, a regular measure of risk $\mathcal{E}$ is obtained by:*

$$\mathcal{R}(X) = \min_{C \in \mathbb{R}}\{C + \mathcal{V}(X - C)\} .$$

*If $\mathcal{V}$ is positively homogeneous, $\mathcal{R}$ is positively homogeneous. If $\mathcal{V}$ is monotonic, $\mathcal{R}$ is monotonic.*

(d) *For any regular measure of error $\mathcal{E}$, a regular measure of deviation $\mathcal{D}$ is obtained by*

$$\mathcal{D}(X) = \min_{C \in \mathbb{R}}\{\mathcal{E}(X - C)\} .$$

*If $\mathcal{E}$ is positively homogeneous, $\mathcal{D}$ is positively homogeneous. If $\mathcal{E}$ satisfies the condition $\mathcal{E}(X) \leq |\mathbb{E}[X]|$ for $X \leq 0$, then $\mathcal{D}$ satisfies the condition $\mathcal{D}(X) \leq \sup X - \mathbb{E}[X]$ for all $X$.*

(e) *In both (c) and (d), as long as the expression being minimized is finite for some $C$, the set of $C$ values for which the minimum is attained is a nonempty, closed, bounded interval. Moreover when $\mathcal{V}$ and $\mathcal{E}$ are paired as in (b), the interval comes out the same and gives the associated statistic:*

$$\arg\min_{C \in \mathbb{R}}\{C + \mathcal{V}(X - C)\} = \mathcal{S}(X) = \arg\min_{C \in \mathbb{R}}\{\mathcal{E}(X - C)\} .$$

**Remark A.1** (Error Projection and Certainty Equivalence). In order to establish the validity of a given quartet $(\mathcal{R}, \mathcal{D}, \mathcal{V}, \mathcal{E})$ as a quadrangle, it is sufficient to demonstrate the satisfaction of either conditions (Q1) and (Q3), or conditions (Q2) and (Q3), as conditions (Q1) and (Q2) are intrinsically linked through the condition (Q3). Indeed, $\mathcal{R}(X) = \inf_C \big\{ C + \mathcal{V}(X - C) \big\} = \inf_C \big\{ \mathcal{E}(X - C) \big\} + \mathbb{E}[X] = \mathcal{D}(X) + \mathbb{E}[X]$.

# B    FUNCTIONAL SPACE SETTING

This subsection discusses the logic behind choosing $\mathcal{L}^2$ as a working space. The choice of $\mathcal{L}^p := \mathcal{L}^p(\Omega, \Sigma, P_0)$, $p \in [1, \infty)$ seems to be reasonable, however, one still has to be careful since if $\mathcal{R} : \mathcal{L}^p \to \overline{\mathbb{R}}$ is a proper convex risk measure, then either $\mathcal{R}(\cdot)$ is finite valued and continuous on $\mathcal{L}^p$ or $\mathcal{R}(X) = +\infty$ on a dense set of points $X \in \mathcal{L}^p$ (cf. (Shapiro et al., 2014, Proposition 6.8)). Therefore, for some risk measures, it may be even impossible to find an appropriate space.

For $\varphi$-divergence risk measures, the natural choice of a functional space can be an Orlicz space paired with a divergence function satisfying $\varphi(0) < +\infty$, $\lim_{x \to +\infty} \varphi(x)/x = +\infty$, suggested by Dommel and Pichler (2020) and adopted by Fröhlich and Williamson (2022b).

However, this particular space excludes important divergence functions such as the total variation distance (TVD). The TVD fits in the framework of Shapiro (2017), which uses $\mathcal{L}^p$ in general and switches to $\mathcal{L}^\infty$ for certain divergence functions. Of course, the simplest way would be to work with finite $\Omega$. Then every function $X : \Omega \to \mathbb{R}$ is measurable, and the space of all such functions can be identified with the Euclidean space. Such an approach was taken by Bayraksan and Love (2015).

In light of everything mentioned above, we follow (Rockafellar and Uryasev, 2013) and take $\mathcal{L}^2$ as our working space assuming finiteness where needed. This choice also allows us to rely on the extensive theory developed for the FRQ in this setting.

# C    PROOF OF THEOREM 3.1

*Proof.* First, we verify the conditions for regular risk measure in Definition 2.4.

*Closedness and Convexity*: Since the envelope $\mathcal{Q}$ is closed and convex ((Rockafellar et al., 2006; Rockafellar and Uryasev, 2013)), then $\mathcal{R}_{\varphi,\beta}(X)$ is closed (lower semicontinuous) and convex as a maximum of continuous affine functions.

*Constancy*: Constancy is implied by the condition $\mathbb{E}Q = 1$,

$$\sup_{Q \in \mathcal{Q}^1_{\varphi,\beta}} \mathbb{E}[CQ] = \sup_{Q \in \mathcal{Q}^1_{\varphi,\beta}} C\,\mathbb{E}[Q] = C \ .$$

*Risk aversity*: We can construct a $Q_0$ such that the strict inequality holds for $\mathcal{R}_{\varphi,\beta}(X) > \mathbb{E}[X]$. As a function of $r$, $P(X \le r)$ is a nondecreasing, right-continuous function with a range in $[0, 1]$. Thus for a nonconstant $X$, there exists $r \in \mathbb{R}$, $p \in (0, 1)$ such that $P(X \le r) = p$, $P(X > r) = 1 - p$. By convexity of $\varphi(x)$ and $1 \subset \text{int}(\{x : \varphi(x) < +\infty\})$, there exists $\delta > 0$ such that $\varphi(x) \le \beta$ for $x \in (1 - \delta, 1 + \delta)$. Then, there exists $\delta_1 \in (0, \delta)$, $\delta_2 \in (0, \delta)$ such that $\delta_1 = \frac{1-p}{p}\delta_2$. Define $Q_0$ by

$$Q_0(\omega) = \begin{cases} 1 - \delta_1, & \omega : X(\omega) \le r \\ 1 + \delta_2, & \omega : X(\omega) > r \end{cases} . \tag{C.1}$$

The feasibility can be checked by $\mathbb{E}[\varphi(Q_0)] \le \beta$, $\mathbb{E}Q_0 = 1$.

We have

$$\mathbb{E}[XQ_0] = \mathbb{E}[XQ_0 | X \le r]P(X \le r) + \mathbb{E}[XQ_0 | X > r]P(X > r) \tag{C.2}$$

$$= p(1 - \delta_1)\mathbb{E}[X | X \le r] + (1 - p)(1 + \delta_2)\mathbb{E}[X | X > r] \tag{C.3}$$

$$= p\mathbb{E}[X | X \le r] + (1 - p)\mathbb{E}[X | X > r] - p\delta_1\mathbb{E}[X | X \le r] + (1 - p)\delta_2\mathbb{E}[X | X > r] \tag{C.4}$$

$$= \mathbb{E}[X] + p\delta_1(\mathbb{E}[X | X > r] - \mathbb{E}[X | X \le r]) \tag{C.5}$$

$$> \mathbb{E}[X] \ . \tag{C.6}$$

Thus $\mathcal{R}_{\varphi,\beta}(X)$ is a regular risk measure.

Next, we verify the conditions for regular regret measure.

*Closedness and Convexity:* Same with the proof above for regular risk measure.

*Risk aversity:* For $X \neq const$,

$$\mathcal{V}_{\beta,\varphi}(X) \geq \mathcal{R}_{\beta,\varphi}(X) > \mathbb{E}X. \tag{C.7}$$

The first inequality is due to $\mathcal{Q}^1_{\varphi,\beta} \subset \mathcal{Q}_{\varphi,\beta}$.

*Zeroness:*

$$\sup_{Q \in \mathcal{Q}_{\varphi,\beta}} \mathbb{E}[0 \cdot Q] = 0. \tag{C.8}$$

The proof of Theorem 1 in Sun et al. (2020) (which works on coherent risk measure) can be applied here to show that a regular regret measure can be obtained by removing condition $\mathbb{E}Q = 1$ in 3.6. Thus the risk 3.1 and regret 3.2 satisfies (Q2) in Definition 2.8.

Deviation 3.3 and error 3.4 measure are obtained by centerness formulae (Q3) (see Definition 2.8). With Theorem A.1, we can show the regularity of deviation and error, and that the minimum in $C$ for a regular regret measure is attainable. The optimal $C$ is $\mathcal{S}_{\varphi,\beta}(X)$. $\qquad\square$

The proof of aversity of the risk measure constructs a feasible random variable inspired by Ang et al. (2018). The proof of the relation between risk and regret follows Sun et al. (2020). Ang et al. (2018); Sun et al. (2020) work with coherent risk measures. The proving techniques are of broader interest. Ang et al. (2018) proves that 1 being a relative interior point of the envelope $\mathcal{Q}$ is sufficient for a coherent risk measure to be risk averse. Sun et al. (2020) proves that removing $\mathbb{E}Q = 1$ in the envelope of coherent risk measure generates a coherent regret measure.

Proposition 4.1 of Artzner et al. (1999) proves the coherency of risk measures that have representation $\sup_{P \in \mathcal{P}} \mathbb{E}_P[X]$ for any set $\mathcal{P}$. The setting in Artzner et al. (1999) is finite $\mathcal{R}(X)$ and finite $\Omega$.

An alternative proof of risk 3.1 and regret 3.2 satisfying (Q2) in Definition 2.8 can be obtained from the primal representations in Section 3.3. The relation (Q2) can be directly observed from the primal risk and regret.

## D    PROOF OF THEOREM 3.2

*Proof.* Consider the regret 3.2

$$\mathcal{V}_{\varphi,\beta} = \sup_{Q \in \mathcal{Q}_{\varphi,\beta}} \mathbb{E}[XQ] = -\inf_{Q \in \mathcal{Q}_{\varphi,\beta}} \mathbb{E}[-QX]. \tag{D.1}$$

Consider the Lagrangian dual problem of $\inf_{Q:Q \in \mathcal{Q}_{\varphi,\beta}} \mathbb{E}[-QX]$

$$\sup_{t \geq 0} \inf_{Q} \left\{ \mathbb{E}[-XQ] + t \left( \mathbb{E}[\varphi(Q)] - \beta \right) \right\}. \tag{D.2}$$

Denote the optimal $t$ by $t^*$. If $t^* = 0$, then

$$\sup_{t \geq 0} \inf_{Q} \left\{ \mathbb{E}[-XQ] + t \left( \mathbb{E}[\varphi(Q)] - \beta \right) \right\} = \inf_{Q} \mathbb{E}[-XQ] = -\infty. \tag{D.3}$$

Thus for all $t \geq 0$,

$$\inf_{Q} \left\{ \mathbb{E}[-XQ] + t \left( \mathbb{E}[\varphi(Q)] - \beta \right) \right\} = -\infty . \tag{D.4}$$

Thus if $t^* = 0$, the optimum is also attained at $t > 0$. If $t^* > 0$, $t > 0$ and $t \geq 0$ are the same for the problem. Thus, we can substitute $t \geq 0$ with $t > 0$ in the Lagrange dual problem.

Then,

$$\sup_{t>0} \inf_Q \{\mathbb{E}[-XQ] + t\left(\mathbb{E}[\varphi(Q)] - \beta\right)\} \tag{D.5}$$

$$= \sup_{t>0} \inf_Q (-t) \left\{\mathbb{E}\left[\frac{X}{t}Q - \varphi(Q)\right] + \beta\right\} \tag{D.6}$$

$$= - \inf_{t>0} \sup_Q t \left\{\mathbb{E}\left[\frac{X}{t}Q - \varphi(Q)\right] + \beta\right\} . \tag{D.7}$$

Next, we prove that

$$- \inf_{t>0} \sup_Q t \left\{\mathbb{E}\left[\frac{X}{t}Q - \varphi(Q)\right] + \beta\right\} = - \inf_{t>0} t \left\{\beta + \mathbb{E}\varphi^*\left(\frac{X}{t}\right)\right\}. \tag{D.8}$$

We consider two cases where the following condition is satisfied and not satisfied

$$\sup_Q \left\{\mathbb{E}\left[\frac{X}{t}Q - \varphi(Q)\right]\right\} < +\infty \quad \text{for some } t . \tag{D.9}$$

When D.9 is satisfied, since $XQ/t - \varphi(Q)$ is a normal convex integrand (Shapiro, 2017), $sup$ and expectation in D.7 are exchangeable by Theorem 3A of Rockafellar (1976). Thus, D.8 holds.

When D.9 is not satisfied, $\sup_Q \{\mathbb{E}[XQ/t - \varphi(Q)]\} = +\infty$ for all $t$. We have

$$- \inf_{t>0} \sup_Q t \left\{\mathbb{E}\left[\left(\frac{X}{t}\right)Q - \varphi(Q)\right] + \beta\right\} = -\infty.$$

We also have that

$$t\left(\mathbb{E}\varphi^*\left(\frac{X}{t}\right) + \beta\right) = t\left(\mathbb{E}\left[\sup_Q \left\{\left(\frac{X}{t}\right)Q - \varphi(Q)\right\}\right] + \beta\right) \tag{D.10}$$

$$\geq \sup_Q t \left\{\mathbb{E}\left[\left(\frac{X}{t}\right)Q - \varphi(Q)\right] + \beta\right\} \tag{D.11}$$

$$= +\infty. \tag{D.12}$$

Thus

$$- \inf_{t>0} t \left(\mathbb{E}\varphi^*\left(\frac{X}{t}\right) + \beta\right) = -\infty. \tag{D.13}$$

We see that D.8 holds with or without the condition D.9. With D.5–D.7, D.8, we obtain

$$\sup_{t>0} \inf_Q \{\mathbb{E}[-XQ] + t\left(\mathbb{E}[\varphi(Q)] - \beta\right)\} = - \inf_{t>0} t \left\{\beta + \mathbb{E}\varphi^*\left(\frac{X}{t}\right)\right\} . \tag{D.14}$$

Strong duality for the convex problem holds since the following Slater's condition is valid for $Q = 1$

$$\exists Q \ : \ Q \in \mathcal{Q}_{\varphi,\beta}, \ \mathbb{E}[\varphi(Q)] < \beta . \tag{D.15}$$

Thus

$$\mathcal{V}_{\varphi,\beta} = - \inf_{Q \in \mathcal{Q}_{\varphi,\beta}} \mathbb{E}[-QX] = \inf_{t>0} t \left\{\beta + \mathbb{E}\varphi^*\left(\frac{X}{t}\right)\right\} . \tag{D.16}$$

By regularity, the statistic $S_{\varphi,\beta}(X)$ is attainable. Denote the optimal $C$ and $t$ by $C^*$ and $t^*$. If $t^* > 0$, $\frac{S_{\varphi,\beta}(X)}{t^*}$ is attainable. We showed that if $t^* = 0$, any $t > 0$ is also optimal. $\frac{S_{\varphi,\beta}(X)}{t^*}$ is attainable. By change of variable, $C^*$ in 3.7,3.8 equals $\frac{S_{\varphi,\beta}(X)}{t^*}$. Thus $t^*$ and $C^*$ in 3.7–3.11 are attainable.

The primal representation of the other elements can be obtained similarly by Lagrange dual problem, or by direct calculation using the quadrangle relations in Definition 2.8. □

The primal representation of the risk measure 3.7 has been studied in the literature under different technical conditions. Fröhlich and Williamson (2022b) starts with the primal representation of coherent regret and obtains the coherent risk with (Q3) centerness relation in Definition 2.8.

# E    MORE EXAMPLES

**Example 9** (Expectile Quadrangle Generated by Generalized Pearson $\chi^2$-divergence)**.** Let $0 < p < 1$. Consider the following extended divergence function and its convex conjugate

$$\varphi(x) = \begin{cases} \frac{1}{q}(x-1)^2, & x > 1 \\ \frac{1}{1-q}(x-1)^2, & x \leq 1 \end{cases} \text{ and } \varphi^*(z) = \begin{cases} \frac{qz^2}{4} + z, & z > 0 \\ \frac{(1-q)z^2}{4} + z, & z \leq 0 \end{cases}. \tag{E.1}$$

The complete quadrangle is as follows

$$\mathcal{R}_{\varphi,\beta}(X) = q\mathbb{E}[(((X - e_q(X))_+)^2] + (1-q)\mathbb{E}[(((X - e_q(X))_-)^2] + \mathbb{E}[X]$$

$$\mathcal{V}_{\varphi,\beta}(X) = \mathbb{E}[X] + \sqrt{\beta E\left[qX_-^2 + (1-q)X_+^2\right]}$$

$$\mathcal{D}_{\varphi,\beta}(X) = q\mathbb{E}[(((X - e_q(X))_+)^2] + (1-q)\mathbb{E}[(((X - e_q(X))_-)^2]$$

$$\mathcal{E}_{\varphi,\beta}(X) = \sqrt{\beta E\left[qX_-^2 + (1-q)X_+^2\right]} = \text{ asymmetric squared loss, scaled}$$

$$\mathcal{S}_{\varphi,\beta}(X) = e_q(X) = \text{ expectile}$$

We recover one version of expectile quadrangle in Malandii et al. (2024). The divergence function $\varphi(x)$ gives rise to a generalized Pearson $\chi^2$-divergence. Example 2 is a special case of this quadrangle with $q = 0.5$.

**Example 10** (Example Generated by finite-interval-indicator Divergence)**.** Let $0 < a < 1 < b$. The divergence function and its convex conjugate are

$$\varphi(x) = \begin{cases} +\infty, & x \in [0, a) \\ 0, & x \in [a, b] \\ +\infty, & x \in (b, +\infty) \end{cases}, \quad \varphi^*(z) = \begin{cases} az, & z < 0 \\ bz, & z \geq 0 \end{cases}. \tag{E.2}$$

The error measure is

$$\mathcal{E}_{\varphi,\beta}(X) = \mathbb{E}[(1-a)X_- + (b-1)X_+]. \tag{E.3}$$

The complete quadrangle is

$$\mathcal{R}_{\varphi,\beta}(X) = (1-a)\text{CVaR}_{\frac{b-1}{b-a}}(X) + a\mathbb{E}[X], \qquad \mathcal{V}_{\varphi,\beta}(X) = \mathbb{E}[(2-a)X_- + bX_+],$$

$$\mathcal{D}_{\varphi,\beta}(X) = (1-a)\text{CVaR}_{\frac{b-1}{b-a}}(X) + (a-1)\mathbb{E}[X], \quad \mathcal{E}_{\varphi,\beta}(X) = \mathbb{E}[(1-a)X_- + (b-1)X_+],$$

$$\mathcal{S}_{\varphi,\beta}(X) = \arg\min_{C \in \mathbb{R}} \mathbb{E}[(1-a)(X-C)_- + (b-1)(X-C)_+],$$

The risk measure in this quadrangle is studied in Pflug and Ruszczynski (2004), Ben-Tal and Teboulle (2007) (see Example 2.3), Love and Bayraksan (2015) (see Example 3). CVaR is a special case of this risk measure for $a = 0$. When $\alpha/(1-\alpha) = (b-1)/(1-a)$, the quadrangle is a scaled version of Example 3.

The risk measure provides another way to connect expectile $e_q(X)$ with distributionally robust optimization (see Proposition 9 in Bellini et al. (2014))

$$e_q(X) = \max_{\gamma \in [\frac{1-q}{q}, 1]} \mathcal{R}_{I_{[\gamma, \gamma\frac{q}{1-q}]}, \beta}(X). \tag{E.4}$$

# F    DERIVATION FOR EXAMPLES

This section contains derivations of the examples in Section 4.

## F.1    EXAMPLE 1

The regret measure is given by

$$\mathcal{V}_{\varphi,\beta}(X) = \inf_{\substack{t > 0 \\ t \geq -\text{ess inf } X \\ t \geq \text{ess sup } X}} \{t\beta + \mathbb{E}[X]\} \tag{F.1}$$

$$= \beta \max\{0, -\text{ess inf } X, \text{ess sup } X\} + \mathbb{E}[X] \tag{F.2}$$

$$= \beta \text{ ess sup } |X| + \mathbb{E}[X]. \tag{F.3}$$

The risk measure is given by

$$\mathcal{R}_{\varphi,\beta}(X) = \inf_{\substack{t>0,\, C\in\mathbb{R} \\ t(C-1)\leq \operatorname{ess\,inf} X \\ t(C+1)\geq \operatorname{ess\,sup} X}} \{t\beta + tC + \mathbb{E}[X - tC]\} \tag{F.4}$$

$$= \frac{\beta}{2}(\operatorname{ess\,sup} X - \operatorname{ess\,inf} X) + \mathbb{E}[X] . \tag{F.5}$$

From the constraints $t(C - 1) \leq \operatorname{ess\,inf} X$ and $t(C + 1) \geq \operatorname{ess\,sup} X$, we have

$$2t \geq \operatorname{ess\,sup} X - \operatorname{ess\,inf} X,$$

hence the optimal

$$t^* = (\operatorname{ess\,sup} X - \operatorname{ess\,inf} X)/2.$$

From the constraints, we have

$$2\operatorname{ess\,sup} X/(\operatorname{ess\,sup} X - \operatorname{ess\,inf} X) - 1 \leq C \leq 2\operatorname{ess\,inf} X/(\operatorname{ess\,sup} X - \operatorname{ess\,inf} X) + 1.$$

Thus,

$$(\operatorname{ess\,sup} X + \operatorname{ess\,inf} X)/(\operatorname{ess\,sup} X - \operatorname{ess\,inf} X) \geq C \geq (\operatorname{ess\,sup} X + \operatorname{ess\,inf} X)/(\operatorname{ess\,sup} X - \operatorname{ess\,inf} X),$$

yelding

$$C^* = (\operatorname{ess\,sup} X + \operatorname{ess\,inf} X)/(\operatorname{ess\,sup} X - \operatorname{ess\,inf} X).$$

Therefore, the statistic

$$\mathcal{S}_{\varphi,\beta} = C^* t^* = (\operatorname{ess\,sup} X + \operatorname{ess\,inf} X)/2.$$

## F.2 EXAMPLE 4

The equation for the statistic can be obtained from the second equation in H.9 when $\varphi^*(z) = \exp(z) - 1$.

## F.3 EXAMPLE 5

The risk measure is given by

$$\mathcal{R}_{\varphi,\beta}(X) = \inf_{\substack{t>0,\, C\in\mathbb{R} \\ \operatorname{ess\,sup}(X-C)\leq t}} \{t\beta + C - t + \mathbb{E}[X - C + t]_+\}$$

$$= \inf_{\substack{t>0,\, C\in\mathbb{R} \\ \operatorname{ess\,sup}(X-C-t)\leq t}} \{t\beta + C + \mathbb{E}[X - C]_+\}$$

$$= \inf_{\substack{t>0,\, C\in\mathbb{R} \\ \operatorname{ess\,sup}(X)-2t\leq C}} \{t\beta + C + \mathbb{E}[X - C]_+\} .$$

The function being minimized is convex in $C$. It attains minimum at $C \in (-\infty, \operatorname{ess\,inf} X]$ if there is no constraint on $C$. Thus the minimum in $C$ is attained at $C^* = \operatorname{ess\,sup}(X) - 2t$. Suppose that $\beta \in (0, 2)$ (Note that TVD is no larger than 2). Then

$$\mathcal{R}_{\varphi,\beta}(X) = \operatorname{ess\,sup}(X) + \inf_{t>0} \{t(\beta - 2) + \mathbb{E}[X - \operatorname{ess\,sup}(X) + 2t]_+\}$$

$$= \operatorname{ess\,sup}(X) + \inf_{t<0} \left\{t(1 - \frac{\beta}{2}) + \mathbb{E}[X - \operatorname{ess\,sup}(X) - t]_+\right\}$$

$$= \operatorname{ess\,sup}(X) + (1 - \frac{\beta}{2}) \inf_{t<0} \left\{t + (1 - \frac{\beta}{2})^{-1}\mathbb{E}[X - \operatorname{ess\,sup}(X) - t]_+\right\} .$$

Note that since $X - \operatorname{ess\,sup}(X) \leq 0$, the minimum in the last equation is attained at some $t \leq 0$, and this minimum is equal to

$$\operatorname{CVaR}_{\frac{\beta}{2}}(X - \operatorname{ess\,sup}(X)) = \operatorname{CVaR}_{\frac{\beta}{2}}(X) - \operatorname{ess\,sup}(X).$$

### F.4    EXAMPLE 2

The extended Pearson $\chi^2$-divergence risk measure is given by

$$\mathcal{R}_{\varphi,\beta}(X) = \inf_{t>0, C\in\mathbb{R}} t \left\{ C + \beta + \frac{1}{4t^2}\mathbb{E}[(X-C)^2] + \mathbb{E}[\frac{X-C}{t}] \right\}$$

$$= \inf_{t>0, C\in\mathbb{R}} \left\{ t\beta + \frac{1}{4t}\mathbb{E}[(X-C)^2] + \mathbb{E}[X] \right\}$$

$$= \mathbb{E}[X] + \sqrt{\beta\mathbb{V}[X]},$$

where $\mathbb{V}[X] = \mathbb{E}[(X-\mathbb{E}[X])^2]$ is the variance of $X$ and $(t^*, C^*)$, which furnish the minimum are

$$t^* = \sqrt{\frac{\mathbb{V}[X]}{4\beta}}, \qquad C^* = \mathbb{E}[X].$$

Evidently, the corresponding regret is given by

$$\mathcal{V}_{\varphi,\beta}(X) = \mathbb{E}[X] + \sqrt{\beta\mathbb{E}[X^2]}$$

$$= \mathbb{E}[X] + \sqrt{\beta}\,\|X\|_2\,.$$

### F.5    EXAMPLE 9

The error measure is given by

$$\mathcal{E}_{\varphi,\beta}(X) = \inf_{t>0} t\beta + \mathbb{E}\left[ t\varphi^*\left(\frac{X}{t}\right) - X \right] \tag{F.6}$$

$$= \inf_{t>0} t\beta + \frac{1}{4t}E\left[ qX_-^2 + (1-q)X_+^2 \right] \tag{F.7}$$

$$= t\beta + \frac{1}{4t}E\left[ qX_-^2 + (1-q)X_+^2 \right] \Big|_{t=\sqrt{\frac{E\left[qX_-^2 + (1-q)X_+^2\right]}{4\beta}}} \tag{F.8}$$

$$= \sqrt{\beta\mathbb{E}\left[ qX_-^2 + (1-q)X_+^2 \right]} \tag{F.9}$$

### F.6    EXAMPLE 6

Plugging in the convex conjugate, the regret measure is given by

$$\mathcal{V}_{\varphi,\beta}(X) = \inf_{t>0} t\beta + t\mathbb{E}\left[ \left( \frac{1}{4}\left(\frac{X}{t}\right)^2 + \frac{X}{t} \right) I_{\{\frac{X}{t}\geq-2\}} \right] \tag{F.10}$$

$$= \inf_{t>0} t\beta + \frac{1}{4t}\mathbb{E}\left[ \left((X+2t)^2 - 4t^2\right) I_{\{X+2t>0\}} \right]. \tag{F.11}$$

For the risk measure, we use an equivalent divergence function and its convex conjugate

$$\varphi(x) = \begin{cases} x^2 - 1, & x \geq 0 \\ +\infty, & x < 0 \end{cases}, \quad \varphi^*(z) = \begin{cases} \frac{z^2}{4} + 1, & z \geq 0 \\ 1, & z < 0 \end{cases} = 1 + \frac{z^2}{4}I_{z\geq0}\,. \tag{F.12}$$

Plugging in the convex conjugate, the risk measure is given by

$$\mathcal{R}_{\varphi,\beta}(X) = \inf_{C\in\mathbb{R}, t>0} C + t\beta + t\mathbb{E}\left[ \left( \frac{1}{4}\left(\frac{X-C}{t}\right)^2 I_{\{\frac{X-C}{t}\geq0\}} + 1 \right) \right] \tag{F.13}$$

$$= \min_{C\in\mathbb{R}} \inf_{t>0} C + t(\beta+1) + \frac{1}{4t}\mathbb{E}\left[ (X-C)^2 I_{\{X-C>0\}} \right] \tag{F.14}$$

$$= \min_{C\in\mathbb{R}} C + \sqrt{(\beta+1)\mathbb{E}\left[ (X-C)^2 I_{\{X-C>0\}} \right]}. \tag{F.15}$$

The optimal $t^*$ is $\sqrt{\frac{\mathbb{E}[(X-C)^2 I_{\{X-C>0\}}]}{4(\beta+1)}}$. By Krokhmal (2007), the optimal $C^*$ is the second-order quantile.

## G    PROOF OF EQUIVALENCE

The equivalence between 5.7 and 5.8 is proved as follows. By Theorem 2.1, Problem 5.7 is equivalent to

$$\min_f \ \mathcal{D}_{\varphi,\beta}(\bar{Z}_f), \qquad \text{calculate } C = \mathcal{S}(\bar{Z}_f) \,. \tag{G.1}$$

The equivalence to Problem 5.8 follows from the dual representation of $\mathcal{D}_{\varphi,\beta}(X)$ in 3.3.

## H    STATISTIC AND RISK IDENTIFIER

### H.1    PROOF OF PROPOSITION J.1

**Lemma H.1** (Convexity). *Let $f : \mathbb{R} \times (0,\infty) \to \mathbb{R}$ be such that*

$$f(C,t) = C + t\beta + \mathbb{E}\Big[t\varphi^*\Big(\frac{X-C}{t}\Big)\Big]. \tag{H.1}$$

*Then $f(C,t)$ is convex in $(C,t)$ and*

$$\partial_{(C,t)}(f(C,t)) = (1,\beta)^\top + \mathbb{E}\left[\partial_{(C,t)}\left(t\varphi^*\left(\frac{X-C}{t}\right)\right)\right], \tag{H.2}$$

*where $\partial_{(C,t)}(f(C,t))$ denotes a subdifferential of a convex function $f(C,t)$ with respect to the vector $(C,t)^\top \in \mathbb{R} \times (0,\infty)$, cf. (Rockafellar, 1970, Definition 23.1). The "+" sign in H.2 is understood in the sense of the Minkowski sum.*

*Proof.* To prove the first part of the lemma it suffices to establish that the function

$$\psi(z,t) = t\varphi^*(z/t), \quad z \in \mathbb{R}, \ t \in (0,\infty)$$

is convex. This follows from the fact that the function $h(z,t) = tg(z/t)$, $z \in \mathbb{R}^n$, $t > 0$ is convex if and only if $g$ is convex. Such function $h$ is called a *perspective function,* cf. (Dacorogna and Maréchal, 2008, Lemma 2.1). Hence, since $\varphi^*$ is convex then $\psi$ is also convex as a perspective function. Therefore, $f(C,t)$ is convex since convexity is preserved under linear transformations.

The second part of the lemma follows from (Rockafellar, 1977, Theorem 23). Indeed, since the function under the expectation in H.1 is convex, hence measurable (cf. Rockafellar and Wets (1998)), the subdifferential can be interchanged with the expectation. □

**Proposition H.1** (Proposition J.1). *Let $(\mathcal{R}_{\varphi,\beta}, \mathcal{D}_{\varphi,\beta}, \mathcal{V}_{\varphi,\beta}, \mathcal{E}_{\varphi,\beta})$ be a primal extended $\varphi$-divergence quadrangle. Statistic in this quadrangle equals*

$$\mathcal{S}_{\varphi,\beta}(X) = \left\{C \in \mathbb{R} : 0 \in (1,\beta)^\top + \mathbb{E}\left[\partial_{(C,t)}\left(t\varphi^*\left(\frac{X-C}{t}\right)\right)\right]\right\}. \tag{H.3}$$

*Proof.* Definition 2.8 implies that the statistic is equal to

$$\begin{aligned} \mathcal{S}_{\varphi,\beta}(X) &= \arg\min_{C \in \mathbb{R}}\big\{ C + \mathcal{V}_{\varphi,\beta}(X - C) \big\} \\ &= \arg\min_{C \in \mathbb{R}} \inf_{t>0} f(C,t) \,, \end{aligned} \tag{H.4}$$

where $f(C,t) = C + t\beta + \mathbb{E}\big[t\varphi^*\big(\frac{X-C}{t}\big)\big]$. To find the statistic one has to minimize $f(C,t)$ with respect to $(C,t)$. Since $f(C,t)$ is convex, cf. Lemma H.1, then it reaches the minimum if and only if

$$0 \in \partial_{(C,t)} f(C,t) \,. \tag{H.5}$$

Therefore, cf. Lemma H.1, condition H.5 is equivalent to

$$0 \in (1,\beta)^\top + \mathbb{E}\left[\partial_{(C,t)}\left(t\varphi^*\left(\frac{X-C}{t}\right)\right)\right]. \tag{H.6}$$

□

If for an extended divergence function $\varphi(x)$, the conjugate $\varphi^*(z)$ is positive homogeneous, then the expression 3.11 for statistic is reduced to

$$\mathcal{S}_{\varphi,\beta}(X) = \arg\min_{C \in \mathbb{R}} \; C + \mathbb{E}[\varphi^*(X - C)] \; . \tag{H.7}$$

The (Rockafellar and Uryasev, 2013, Expectation Theorem) in this case implies

$$\mathcal{S}_{\varphi,\beta}(X) = \left\{ C \in \mathbb{R} : \mathbb{E}\left[ \left. \frac{\partial^-}{\partial z} \varphi^*(z) \right|_{x = X - C} \right] \le 1 \le \mathbb{E}\left[ \left. \frac{\partial^+}{\partial z} \varphi^*(z) \right|_{x = X - C} \right] \right\}, \tag{H.8}$$

where $\dfrac{\partial^-}{\partial z}, \dfrac{\partial^+}{\partial z}$ denote left and right derivatives with respect to $z \in \mathbb{R}$. As a finite convex homogeneous function, $\varphi^*(z)$ is the support function of a closed interval (Corollary 13.2.2, Rockafellar (1970)). The convex conjugate of a support function is an indicator function. Since $\varphi(1) = 0$, it must be in the form of the $\varphi(x)$ in Example 10.

In fact, Dommel and Pichler (2020) provided optimality conditions for $(C, t)$ in 3.7. For differentiable function $\varphi^*$, they developed a set of equations known as the *characterizing equations* for optimal $(C, t)$. Further, we provide a system of equations similar to the characterizing equations developed by Dommel and Pichler (2020).

**Definition H.1** (Characterizing Equations). Let $\varphi^*(z) \in C^1(\mathbb{R})$. Characterizing system of equations is defined by:

$$\begin{cases} \mathbb{E}\left[ \left. \dfrac{d\varphi^*(z)}{dz} \right|_{z = \frac{X - C}{t}} \right] = 1 \; , \\[4mm] \beta + \mathbb{E}\left[ \varphi^*\left( \dfrac{X - C}{t} \right) \right] - \dfrac{1}{t}\mathbb{E}\left[ (X - C) \left. \dfrac{d\varphi^*(z)}{dz} \right|_{z = \frac{X - C}{t}} \right] = 0 \; . \end{cases} \tag{H.9}$$

The following Corollary H.1 provides an expression for the statistic $\mathcal{S}_{\varphi,\beta}$ with smooth $\varphi^*(z)$.

**Corollary H.1** (Characterization of $\mathcal{S}_{\varphi,\beta}$ : Smooth Case). *Let $\varphi^*(z) \in C^1(\mathbb{R})$, then the statistic equals*

$$\mathcal{S}_{\varphi,\beta}(X) = \{ C \in \mathbb{R} : (C, t) \text{ is a solution to Characterizing Equations H.9 } \}.$$

*Proof.* Replacing the subdifferential in H.6 with the gradient $\nabla_{(C,t)}$ leads to the system of equations H.9. $\qquad\square$

## H.2 Proof of Proposition 6.1

**Lemma H.2** (Subgradients of expectation, Bauschke and Combettes (2011)). *Let $(\Omega, \mathcal{A}, P_0)$ be a probability space and $\psi : \mathbb{R} \to \overline{\mathbb{R}}$ be a proper, lsc, and convex function. Set*

$$\rho = \mathbb{E}[\psi(X)]. \tag{H.10}$$

*Then $\rho$ is proper, convex lsc functional and, for every $X \in \mathrm{dom}(\rho)$,*

$$\partial_X \rho(X) = \{ Q \in \mathcal{L}^2 : Q \in \partial\psi(X) \; \mathbb{P}_0 - a.s.\}. \tag{H.11}$$

**Proposition H.2** (Proposition 6.1). *Denote by $C^*$ and $t^*$ the optimal $C$ and $t$ in the primal representation 3.7 of extended $\varphi$-divergence risk measure. The risk identifier of risk measure $\mathcal{R}_{\varphi,\beta}(X)$ can be expressed as follows*

$$Q^*(\omega) \in \partial\varphi^*\left( \frac{X(\omega)}{t^*} - C^* \right). \tag{H.12}$$

*Denote by $C^*$ the optimal $C$ in the primal representation 3.10 of extended $\varphi$-divergence risk measure. The risk identifier of extended $\varphi$-divergence error measure $\mathcal{E}_{\varphi,\beta}(X)$ can be expressed as follows*

$$Q^*(\omega) \in \partial\varphi^*\left( \frac{X(\omega)}{t^*} \right). \tag{H.13}$$

*Proof.* It is known that the risk identifier is the subgradient of the risk function (see, for example, Proposition 8.36 of Royset and Wets (2022)). Therefore, H.12 is obtained by taking the subdifferential of 3.7 following Lemma H.2. The expression H.13 is obtained analogously. □

Note that the envelope $\mathcal{Q}_{\varphi,\beta}$ of error does not have the constraint $\mathbb{E}Q = 1$. However, when we minimize $\mathcal{E}_{\varphi,\beta}(X - C)$ with respect to $C$ to get statistic $\mathcal{S}_{\varphi,\beta}(X)$, the constraint $\mathbb{E}Q = 1$ is satisfied automatically. This can be seen from the necessary condition for saddle point $(C^*, Q^*)$

$$\frac{\partial}{\partial C}\mathbb{E}[(X - C)(Q^* - 1)]\Big|_{C=C^*} = 0. \tag{H.14}$$

# I    PROOF OF PROPOSITION 7.1

**Proposition I.1** (Proposition 7.1). *Let $\varphi(x)$ be a divergence function. $\varphi$-divergence can be recovered from the elements in $\varphi$-divergence quadrangle by*

$$D_\varphi(P||P_0) = \sup_{X \in \mathcal{L}^2, \beta > 0} \{\mathbb{E}[XQ] - \mathcal{R}_{\varphi,\beta}(X) - \beta\} \tag{I.1}$$

$$= \sup_{X \in \mathcal{L}^2, \beta > 0} \{\mathbb{E}[X(Q - 1)] - \mathcal{D}_{\varphi,\beta}(X) - \beta\} \tag{I.2}$$

$$= \sup_{X \in \mathcal{L}^2, \beta > 0, C} \{\mathbb{E}[XQ] - \mathcal{V}_{\varphi,\beta}(X - C) + C - \beta\} \tag{I.3}$$

$$= \sup_{X \in \mathcal{L}^2, \beta > 0, C} \{\mathbb{E}[X(Q - 1)] - \mathcal{E}_{\varphi,\beta}(X - C) - \beta\}. \tag{I.4}$$

*Proof.* From **??**, we have by convex conjugate

$$\mathcal{R}_{t\varphi}(X) = \inf_{\beta > 0}\{t\beta + \mathcal{R}_{\varphi,\beta}(X)\}. \tag{I.5}$$

I.5 is a generalization of Proposition 3.1 in Föllmer and Knispel (2011).

Next, we have

$$\mathbb{E}[\varphi(Q)] = \sup_{X \in \mathcal{L}^2}\{\mathbb{E}[XQ] - \mathcal{R}_\varphi(X)\} \tag{I.6}$$

$$= \sup_{X \in \mathcal{L}^2}\{\mathbb{E}[XQ] - \inf_{\beta > 0}\{\beta + \mathcal{R}_{\varphi,\beta}(X)\}\} \tag{I.7}$$

$$= \sup_{X \in \mathcal{L}^2, \beta > 0}\{\mathbb{E}[XQ] - \mathcal{R}_{\varphi,\beta}(X) - \beta\}, \tag{I.8}$$

where I.6 is by **??**, I.7 is by plugging in I.5 to **??**.

Since $\varphi(x)$ is a divergence function, $\mathbb{E}[\varphi(Q)] = D_\varphi(P||P_0)$. The rest of the proof is by quadrangle relations. □

# J    CHARACTERIZATION OF STATISTICS

**Proposition J.1** (Characterization of $\mathcal{S}_{\varphi,\beta}$). *Let $(\mathcal{R}_{\varphi,\beta}, \mathcal{D}_{\varphi,\beta}, \mathcal{V}_{\varphi,\beta}, \mathcal{E}_{\varphi,\beta})$ be a primal extended $\varphi$-divergence quadrangle. Statistic in this quadrangle equals*

$$\mathcal{S}_{\varphi,\beta}(X) = \left\{C \in \mathbb{R} : 0 \in (1, \beta)^\top + \mathbb{E}\left[\partial_{(C,t)}\left(t\varphi^*\left(\frac{X - C}{t}\right)\right)\right]\right\}.$$

# K    DETAILS OF CASE STUDY

**Risk Identifier**    We first solve the problems 5.4, 5.1 and 5.7 in primal representations. With the optimal solutions, we obtain the random variable $X$ in three problems, respectively. By plugging in $\varphi^*(z) = z^2/2 + 1$ to Proposition 6.1, we obtain the risk identifier $Q^* = (X/t^* - C^*)^2/2 + 1$.

**Data**  The data for portfolio optimization and regression are the same: it is generated by drawing 1,000 samples from a bivariate zero-mean Gaussian distribution. The variance of both random variables is 1, while the covariance is 0.5. The data for classification is generated by two normal distributions with different mean and different covariance matrix. The first has mean $(-0.3, 0)$, while the second has mean $(0.3, 0)$. For both distributions, the variance is 0.05 while the covariance is 0.02. Each class has 100 samples. The value of the risk identifier $Q^*$ is represented through the intensity of color. Darker points have larger values.

**Portfolio Optimization**  We illustrate the idea with Markowitz portfolio optimization from the mean quadrangle (Example 2). The data points $(x, y)$ represents the loss (negative return) of two assets. We choose $\beta = 100$. The optimal portfolio weight is $(0.4999038, 0.5000962)$. The value of the risk identifier $Q^*$ is represented through the intensity of color in Figure 1b. Darker color corresponds to a larger value. Larger values are assigned to data points incurring larger loss, i.e., points whose both coordinates are larger.

**Classification**  We illustrate the idea with large margin distribution machine from the mean quadrangle (Example 2). We choose $\beta = 0.01$ and $\gamma(\boldsymbol{w}) = \|\boldsymbol{w}\|_2^2$. The optimal decision line is $y = -28.826x - 0.486$. The circles represent samples with label 1, while the diamonds represent samples with label $-1$. The value of the risk identifier $Q^*$ is represented through the intensity of color in Figure 1a. A darker spot corresponds to a larger value. Larger values are assigned to data points incurring larger loss (negative margin), i.e., points that are correctly classified and have larger perpendicular distance from the optimal decision line.

**Regression**  We illustrate the idea with least squares regression from the mean quadrangle (Example 2). We choose $\beta = 100$. The regression line is $y = 0.495x - 0.0127$. The value of the risk identifier $Q^*$ is represented through the intensity of color in Figure 1c. A darker spot corresponds to a larger value. Larger values are assigned to data points incurring larger loss, i.e., data points further above the regression line.

