# OpenReview forum: "Risk Quadrangle and Robust Optimization Based on $\varphi$-Divergence"
_ICLR.cc/2025/Conference — Submitted to ICLR 2025_

### Official Review · Reviewer_aJtz · 2024-10-22

**Soundness:** 3
**Presentation:** 2
**Contribution:** 2
**Rating:** 5
**Confidence:** 2

**Summary:**

The paper studies how distributional robust optimization (DRO) can be integrated into the fundamental risk quadrangle (FRQ) framework. It derives a dual and a primal formulation and presents many examples.

**Strengths:**

The paper offers many interesting reformulations for various optimization problems.

**Weaknesses:**

I found the structure of the paper very confusing and hard to follow, and I am afraid that many of its important points are just not coming through. Without any word in the introduction the paper jumps into "demonstrating examples", and the reader is left without motivation until section 1.2 which appears only on the 4th page. Also, sections 7 and 8 show results without discussion or motivation.

There are many typos and weigh sentences, some examples:
- I guess "negative" should not be there for "negative asset returns", or why do you only consider negative ones?
- There is $\lambda$ and $\beta$ in the description of the Mean Quadrangle on page 2, I guess there should be some relationship between the two.
- "A specific case of the extended $\varphi$-divergence quadrangle is called $\varphi$-divergence quadrangle ..."
- "The next theorem proves the dual representation of the extended $\varphi$-divergence quadrangle." while the "extended $\varphi$-divergence quadrangle" is a definition, it needs no proof.

**Questions:**

The equivalence of (1.6) and (1.7) does not look correct to me as the former is independent of $\beta$ while the latter is not. In particular, if you choose $\beta = 0$, then Q = 1 almost surely and the objective value of (1.7) is zero while (1.6) might not be. Could you comment on this, is there anything missing here?

---

> ### Author Response · Authors · 2024-11-25
>
> Thank you for taking the time to review our manuscript and examine the details. We have made significant revisions to improve readability and clarify our contributions. Below are point-by-point responses to your comments and questions.
>
>  - We have completely rewritten the Introduction to present the background, motivation, contributions, and literature review in a logical sequence.
>
>     The two examples have been removed. The revised version starts by connecting DRO to FRQ through coherent risk measures and discusses the natural idea of integrating DRO into FRQ. We then raise the issue of non-coherency in some risk measures, such as the important mean-standard-deviation risk measure, which motivates the introduction of the novel extended $\varphi$-divergence. The paper's contributions are now better summarized in a paragraph on the Main Contributions.
>
>  -  We have reorganized the sections for a more coherent structure.
>
>     In the current version, Sec 3 contains the technical results.  Sec 4 contains concrete examples. Sec 5 contains the interpretation and concrete examples. The sections now flow naturally: Sec 3.1 introduces the extended $\varphi$-divergence risk measure and completes the risk quadrangle for the defined risk measure in dual representation. Sec 3.2 derives the primal representation based on Sec 3.1. Sec 4 provides concrete examples of the extended $\varphi$-divergence quadrangle in primal representation. Sec 5.1 uses the dual representation from Sec 3.2 for a RO/DRO interpretation, and Sec 5.2 presents concrete examples of learning tasks, using the examples from Sec 4 and the interpretations from Sec 5.1.
>
>
>  - We have added non-technical comments after theorem statements to explain their implications.
>
>     For Proposition 6.2 (former 7.2), we write:" Proposition 6.2 allows us to directly calculate the risk identifier (worst-case weight) given the solu- tion to the problem in primal representation. It will be used for calculation in Section 8."
>
>     For Proposition 7.1 (former 8,1), we write:" This study starts with developing new risk measures given a $\varphi$-divergence function. There exists a duality between divergence and risk that allows us to recover the $\varphi$-divergence from the elements of the corresponding $\varphi$-divergence quadrangle."
>
>     Propositions 6.1 and 7.1 are not used in this study. They provide interesting insights into the conditions satisfied by the statistic,  and the duality between $\varphi$-divergence and $\varphi$-divergence risk measure.
>
>
> **Typos and weigh sentences:**
>
>  - The wording was indeed unclear. In the context of risk management, the random variable $X$ represents loss. Therefore, if the random variable of asset return is $R$, we let $X=-R$, which is the negative return. It is not the negative part of the return. Similarly, the random loss $X$ in the classification problem under consideration corresponds to the negative margin.
>
>  - We have removed $\lambda$ and replace it with $\sqrt{\beta}$.
>
>  - We have rewritten the paragraph Main Contributions.
>
>  - We have revised the sentence to provide a more precise description:" The next theorem proves that the  dual representation above satisfies the axioms in Section 2.2."
>
> **Responses to questions:**
>
>  - Before former 1.6 and former 1.7, we wrote:" The following two problems have the same optimal solution $(f, C)$." The optimal objective values may not be equal. The reason is that the error measure in the Mean Quadrangle (Example 2) is $\sqrt{\beta}||X||_2$. As $\beta$ tends to zero, the objective value tends to zero as well.
> To address this confusion, we have added back $\sqrt{\beta}$ and restated the general equivalence in Section 5.
>
>
> We sincerely appreciate  your detailed review. We hope that we have effectively addressed your concerns. We are happy to further improve the paper based on your suggestions.

---

> > ### Comment · Reviewer_aJtz · 2024-11-25
> > **Thank you for the answer, I will keep my score.**
> >
> > Thank you for your detailed feedback on how the reviews helped improved your paper. I feel that the revision of the paper is too significant to get it accepted without further review, hence I keep my score.

---

### Official Review · Reviewer_g4r3 · 2024-10-31

**Soundness:** 2
**Presentation:** 1
**Contribution:** 2
**Rating:** 1
**Confidence:** 3

**Summary:**

This paper introduces an extension of the Fundamental Risk Quadrangle (FRQ), a framework that connects risk management, statistical estimation, and optimization. Within this framework, distributionally robust optimization (DRO) based on φ-divergence aims to minimize the worst-case expected loss, where the maximum is taken over a φ-divergence-defined uncertainty set. The authors present the extended φ-divergence and the extended φ-divergence quadrangle, integrating DRO into the FRQ framework. They derive both primal and dual representations for the quadrangle elements, including risk, deviation, regret, error, and statistic. The dual representation allows for interpreting tasks like classification, portfolio optimization, and regression as forms of robust optimization driven by extended φ-divergence. Meanwhile, the primal representation offers tractable convex formulations for these robust optimization problems. Through examples, the paper demonstrates how common problems—such as least-squares regression, quantile regression, support vector machines, and conditional value-at-risk (CVaR) optimization—fit within this unified framework. A case study is also provided, visualizing the optimal solution in the inner maximization problem of robust optimization.

**Strengths:**

The paper attempts to unify DRO with an existing general stochasric optimization framework (FRQ), which is of theoretical interest.

**Weaknesses:**

The paper is extremely hard to read, mostly because it consists of a sequence of incoherent/not well motivated definitions and results. While I feel that the paper might have merit in terms of the topic it aims to study, I believe the authors should consider (1) restructuring the paper in a major way, making it readable and coherent, and (2) possibly resubmitting this work to a journal or some other venue allowing for longer articles -- it really feels like they tried to stuff as much material as possible in ten pages, with a very poor result in terms of presentation. Here are some more specific comments:

1. In general, the authors should avoid the $a(b)$ notation to mean $a\times b$, and should reserve it to mean "$a$ is a function of $b$"

2. Page 2, when the authors introduce some key concepts, becomes almost unreadable. What do these concepts mean? The authors basically just present a wall of hard-to-read math;

3. Page 3 is also quite hard to read -- it presents too much math without any context;

4. The paper continues in the same style as the previous two points, till the very end.

**Questions:**

Although of minor importance compared to my major concerns outlined above, here are two questions:

1. In the illustrative example on Large Margin Distribution Machine, what is $\sigma(\cdot)$ ? From usage below, I guess it denotes the standard deviation of a random variable, but that's not clear at a first reading;

2. On page 2, talking about linear regression, do the authors mean $\Vert \cdot \Vert$ to be the $L^2$-norm for random variables?

To be clear, I think there's many more such points that need clarification/revision throughout the text, but I think this is best left to a future major restructuring effort to put the paper in better shape.

---

> ### Author Response · Authors · 2024-11-25
>
> Thank you for taking the time to review our manuscript. We acknowledge that the previous version was overly technical and lacked intuitive explanations, which blurred the main message we aimed to convey. We have undertaken major restructuring of the paper to improve its readability and coherence. Below, we provide our detailed responses to your comments and questions.
>
>
>  - The brackets in $Q(w^T L)$ have been deleted to avoid confusion with functions.
>
>  - We have completely rewritten the Introduction to present the background, motivation, contributions, and literature review in a logical sequence. The two examples have been removed.
>
>     The revised Introduction begins by discussing the known result that distributionally robust optimization (DRO) can be viewed as minimizing a coherent risk measure. This connection links DRO to the Fundamental Risk Quadrangle (FRQ). We then discuss the natural idea of integrating DRO into FRQ, highlighting the issue of non-coherency of some important risk measures, such as the mean-standard deviation risk measure. This issue motivated the introduction of the novel extended $\varphi$-divergence. The contributions of the paper are now clearly summarized in the paragraph Main Contributions.
>
>  - A new paragraph has been added to Section 2 to gather the notations. Before defining each axiom of the quadrangle elements in Section 2, we included pedagogical comments to clarify their intuition. To help readers grasp the concept, we provide a concrete example from the Mean Quadrangle for each element. For instance, $\mathbb{E}[X] + \lambda \sigma(X)$ is presented as an important example of a risk measure. We have also added comments to the regression theorem to explain how it connects regression with DRO. The technical discussion previously in Section 2.3 has been moved to the appendix to maintain a clear narrative.
>
>  - Non-technical comments have been added throughout the definitions and  theorems to explain their implications. For example, Definition 3.1 of the extended divergence function is given with an example.  The implications of Theorem 3.1 and 3.2 are explained after the theorem statements.
>
>  - We hope that our message becomes clearer with the examples in Section 4 and 5.2. The theorems can be exemplified  with an im example, the Mean Quadrangle (Example 2) generated by the extended $\chi^2$-divergence.  The extended $\chi^2$-divergence function simply extends the divergence function of $\chi^2$-divergence, $\varphi(x) = (x-1)^2$, to the negative domain.  The risk measure of the quadrangle $\mathcal{R}(X) = \mathcal{E}(X) + \sqrt{\beta}\sigma(X)$ is used as objective functions in large-margin distribution machine and Markowitz portfolio optimization, while the error measure $\mathcal{E}(X) = \lambda||X||_2$ is used as the objective function in least squares regression.  The two measures are connected by quadrangle axiom $\mathcal{R}(X)  = \min_C \mathcal{E}(X-C) + \mathbb{E}(X)$. Through the dual representation, the three problems admit interpretation as robust optimization with extended $\chi^2$-divergence ambiguity set (Example 7).  Furthermore, they can be viewed as  conservative versions of DRO with (non-extended)  $\chi^2$-divergence ambiguity set.
>
>
> Responses to questions:
>
>
>  - Yes, $\sigma$ should have been defined before used. It is the standard deviation.
>
>  - Yes, $||\cdot||_p$ denotes the $\mathcal{L}^p$-norm of a random variable.
>
>
> Thank you  again for your comments and questions. We hope that the major revisions address your concerns regarding readability. Please do not hesitate to share any additional feedback, as we are happy to further improve the paper based on your suggestions.

---

> > ### Comment · Reviewer_g4r3 · 2024-11-25
> >
> > I thank the authors for taking into account my comments, which I hope they will incorporate in the next draft of the paper as they see best fit. However, given the major nature of my concerns, I feel that the paper cannot be accepted without a further iteration of the full review process, which is best left for a future resubmission. Therefore, I keep my score unchanged.

---

### Official Review · Reviewer_tk6q · 2024-10-31

**Soundness:** 2
**Presentation:** 1
**Contribution:** 2
**Rating:** 3
**Confidence:** 3

**Summary:**

The Fundamental Risk Quadrangle (FRQ) is a risk management framework introduced by Rockafellar and Uryasev in 2013. It integrates risk management, statistical estimation, and optimisation, providing a unified approach and broader interpretation of these problems.

By introducing specific quadrangles (i.e. a quartet of risk, deviation, regret, and error measures) based on $\varphi$-divergence, the authors demonstrate how Distributionally Robust Optimization (DRO) can be incorporated into the FRQ framework.

The authors first derive dual representations of the quadrangle elements, providing a robust optimization perspective on certain classification, regression, and portfolio optimization problems. They then develop the primal representations, which offer tractable formulations—specifically as convex optimization problems—of the dual representations.

Finally, the authors provide examples of classical problems that fall within this framework.

**Strengths:**

The FRQ framework provides interesting link between various problems in learning and risk management.
The authors further this link by proposing a unified way of looking at some of those problems.

**Weaknesses:**

I found the paper very hard to read:
- The introduction opens with two extended examples but lacks a pedagogical introduction to the FRQ framework, which may be unfamiliar to the learning community;
- The paper lacks coherence, with many paragraphs consisting of sequences of juxtaposed sentences;
- The purpose/message of the paper is hard to grasp;

The paper's contribution appears limited. The authors propose a general method for incorporating DRO into the FRQ framework using $\varphi$-divergences. However, the three main examples presented in Section 5 have been well-studied in the literature, making it unclear what is novel and what was previously established.

**Questions:**

- The dual representation provides a robust optimization (RO) interpretation of the quadrangles elements. Then the authors link RO with DRO in the last paragraph of Section 3. Could the authors explain more precisely this link? In particular, line 318-319, What do they mean by "$Q$ is the Radon-Nikodym derivate $dP_0/dP$"? In particular, what are the distributions $P$ and $P_0$ in this case? It seems to me that for the condition $\mathbb{E}[\varphi(Q)] \leq \beta$ to be expressed as $D_\varphi(P || P_0) \leq \beta$ we would need Q to be distributed according to $P_0$.

- Are the examples presented in the introduction well-known in the literature? If so, could the authors provide relevant references? (Or at least provide a proof in appendix).

- Are there any problems for which the proposed approach offers new primal/dual formulations?

---

> ### Author Response · Authors · 2024-11-25
>
> Thank you for taking the time to review our manuscript and carefully examine the technical discussion. We have made significant revisions to improve readability and clarify our contributions. Below are point-by-point responses to your comments and questions.
>
>
>  - We have completely rewritten the Introduction. In Section 2, we have added comments and examples to the definitions and theorems to clarify the intuition and implications of the FRQ framework.
>
>     - The two examples in the Introduction have been removed. The revised version starts by connecting DRO to FRQ through coherent risk measures and discusses the natural idea of integrating DRO into FRQ. We then raise the issue of non-coherency of some important risk measures, such as the mean-standard-deviation risk measure, which motivates the introduction of the novel extended $\varphi$-divergence.
>
>
>     - In Section 2, we have added pedagogical explanation to the axioms of the quadrangle elements. To help readers grasp the concept of quadrangle elements, we provide a concrete example from the Mean Quadrangle for each element. For instance, $\mathbb{E}[X] + \lambda \sigma(X)$ is presented as an important example of a risk measure. We have also added comments to the regression theorem on how it connects regression with DRO.
>
>  - We have reorganized the sections for a more coherent structure and included comments in each section to clarify the logical relationships.
> In the updated version, Sec 3 contains the technical results.  Sec 4 contains concrete examples. Sec 5 contains the interpretation and concrete examples.   Each section builds naturally on the previous ones, creating a clearer narrative.
>
>
>   - We have updated the paragraph Main Contributions in the Introduction to clarify our contributions. Since this is a common issue raised by reviewers, we will put it in the general comment.
>
>     - We integrate DRO into FRQ by introducing the extended $\varphi$-divergence and the associated quadrangle. Many interesting connections are built through this framework. For example, the connection between regression and RO/DRO, and the connection between RO and DRO.
>
>
>
>  - We have rewritten Section 4 (formerly Section 5) and added new examples.
>
>     - The updated Section 4.1 presents examples of the extended $\varphi$-divergence quadrangle. The quadrangles are known in their primal representation, but the connection to the extended $\varphi$-divergence has not been established. Therefore, the robust optimization interpretation is novel for these quadrangles.
>
>     - The updated Section 4.2 presents examples of the (non-extended) $\varphi$-divergence quadrangle.  The $\varphi$-divergence risk measures in these quadrangles are well-known, so does the DRO interpretation for them. Our contribution is to complete the risk quadrangle for these  risk measures in the primal representation, which, apart from Example 3 (Quantile Quadrangle), had not been established.
>
>     - For all examples above, the quadrangle establishes a novel connection between regression  and DRO (Section 5).

---

> > ### Comment · Reviewer_tk6q · 2024-11-29
> >
> > Thank you for your detailed rebuttal. Given the significant difference between the proposed revised version from the one we reviewed, I believe that this paper would greatly benefit from another full round of reviews. Hence I will keep my score and encourage the authors to re-submit to a future conference.

---

> ### Author Response · Authors · 2024-11-25
>
> Responses to  questions: (due to a compilation issue, the subscript ${\cdot}_{\varphi, \beta}$ are omitted.)
>
>  - We use separate Sections 3.2 and 3.3 to explain more precisely this link. The idea is elaborated below.
>
>     - First, for the (non-extended) $\varphi$-divergence quadrangle, we  establish a one-to-one correspondence between the probability ambiguity set $\mathcal{P}$ and the risk envelope $\mathcal{Q}$.
>         The conditions  $\varphi(x) = +\infty$ for $x<0$ and $\mathbb{E}[\varphi(Q)] \leq \beta$  imply that $Q \geq 0$ almost surely. Define indicator function $\mathcal{I}_A(x) = 1$ if $x \in A$, and $0$ otherwise.  For every $Q\in\mathcal{Q}$, we can verify that $P_Q(A) = \mathbb{E}[\mathcal{I}_A(\omega) Q(\omega)], A \in \Sigma$ is a probability distributionon $(\Omega, \Sigma)$.
>
>         Consider the constant random variable $Q_0=1$. $Q_0 \in \mathcal{Q}$. For every $P\in \mathcal{P}$, we can verify the definition that $Q$ is the Radon-Nikodym derivative $P_Q/P_{Q_0}$. Due to its uniqueness, every $P\in \mathcal{P}$ has a one-to-one correspondence to a $Q\in\mathcal{Q}$.
>
>     - Next, we show that the (non-extended) $\varphi$-divergence  risk measure $\mathcal{R}$ can be written as
>                 $$\mathcal{R}(X) = \sup_{Q \in \mathcal{Q}^1} \mathbb{E}[XQ].$$
>         This is due to that $\mathbb{E}_{P_0}[XQ] = \mathbb{E}_P[X]$
>
>          and that  $\mathbb{E}[\varphi(Q)] = \mathbb{E}_{P_0}[\varphi(P/P_0)]$, which is the $\varphi$-divergence.
>
>     - Then, we consider the extended $\varphi$-divergence quadrangle. The extended risk measure has the same expression as above, except that $\varphi$ is the extended divergence function. Unlike the non-extended case, $Q$ can take negative values. Having a smaller envelope $\mathcal{Q}$, *the $\varphi$-divergence risk measure is upper bounded by its extended version.*
>
>     - We need an interpretation for $Q$, as well as the minimization problem of the extended $\varphi$-divergence risk measure.
>         $Q$ can be viewed as (potentially negative) weight on samples. The minimization of the extended $\varphi$-divergence risk measure can  be interpreted as a *robust optimization*, where the maximum is over a set of weights.
>
>          - Due to the relation between the $\varphi$-divergence risk measure and its extended version, *when the quadrangle elements are used as objective function, the RO is a more conservative version of the corresponding DRO.* An example well-known in the literature (Theorem 8,2 of [3]) is that the mean-standard deviation risk measure (Example 2) bounds the $\chi^2$-divergence risk measure (Example 6), which is a special case of our result.
>
>          - The conditions $\mathbb{E}[\varphi(Q)]\leq \beta$ and $\mathbb{E}[Q] = 1$ imply that for sufficiently small $\beta$, the value of risk identifier $Q$ cannot be negative. Therefore, *with sufficiently small $\beta$, the $\varphi$-divergence quadrangle becomes equivalent to the extended version.*
>
>      - In summary, RO is a more conservative version of the corresponding DRO. With sufficiently small $\beta$, RO is equivalent to DRO.
>
>
>
>  - The Mean Quadrangle and Quantile Quadrangle are now moved from the Introduction to the Example section.
>
>       - The Mean Quadrangle is known in its primal and dual representation (Example 1 of [1]). However, it was not known that the quadrangle is generated by the extended $\varphi$-divergence function.
>
>       - The Quantile Quadrangle is known in its primal and dual representation (Example 2 of [1]). The risk measure of the quadrangle is known to be DRO with indicator divergence ambiguity set [2]. However, it was not observed that the quantile regression, which minimizes the error measure in this quadrangle, is connected with DRO. This connection is built by the regression theorem in FRQ framework.
>
>  -  Yes. For example,  dual representations for the extended $\varphi$-divergence are new, since the extended divergence is a novel  concept from this study. For the (non-extended) quadrangles generated by KL divergence and TVD, the primal and dual representations were established only for the risk measure, but not deviation,  regret, or error measures.
>
>
>
> Thank you again for your feedback. We hope these revisions have addressed your concerns effectively. Please let us know if further clarifications are needed.
>
>
>
>
> References:
>
> [1] Rockafellar, R. T. and Uryasev, S. (2013). The fundamental risk quadrangle in risk management, optimization and statistical estimation. Surveys in Operations Research and Management Science, 18(1-2):33–53.
>
> [2] Ahmadi-Javid, A. (2012). Entropic value-at-risk: A new coherent risk measure. Journal of Optimization Theory and Applications, 155:1105–1123.
>
>
> [3] Kuhn, D., Shafiee, S., and Wiesemann, W. (2024). Distributionally robust optimization.

---

### Official Review · Reviewer_Bojo · 2024-11-02

**Soundness:** 2
**Presentation:** 1
**Contribution:** 1
**Rating:** 3
**Confidence:** 5

**Summary:**

This paper integrates the $\phi$-divergence distributionally robust optimization into the Fundamental Risk Quadrangle framework and presents the primal and dual representation of different elements in that quadrangle. They demonstrate how common cost functions including classification, regression and portfolio optimization are fit into the framework.

**Strengths:**

The paper provides a quite general connection between one generalized f-divergence DRO and the so-called fundamental risk quadrangle and applies to general cost functions.

**Weaknesses:**

# **Confusing Organization**
The paper’s organization makes it challenging to follow, especially for a theoretically-oriented work. Significant revisions would improve clarity and accessibility for a broader ML audience:

-	**Length, Example-Driven Intro**: The first three pages focus heavily on two examples, with numerous mathematical formulas, but lack emphasis on the paper’s main contribution. The connections between the mean, quantile, and extended $\phi$-divergence quadrangle only become apprante after multiple readings, which detracts from the paper utility.

-	**Overly Technical Sections**: Secs 2 and 3 are highly technical without sufficient explanatory context. Some definitions, such as Defs 2.2., 2.3, 2.4, are only referenced once in Def 2.5 and are not essential to the main context. Moving these, along with Sec 2.3 to Appendix, would better suit a general ML audience.


-	**Lack of Cohesion between Sections**: Many disjointed sections create a fragmented flow. Consider reorganizing the technical results by grouping related content (e.g. combining primal-dual discussions in Secs 3 and 4 and merging Secs 5 and 6 to illustrate concrete cost function examples).

-	**Insufficient Explanation of Theorems**: Each theorem would benefit from non-technical explanations to help readers understand its meaning and implications. Currently, the lack of such interpretations makes it difficult to grasp the practical relevance of the results. For instance, the purpose and utility of Propositions 7.1, 7.2, and 8.1 are unclear from a practical standpoint—why and when would these results matter?

# Unclear Contribution
The paper’s contributions, particularly in the examples and novel interpretations, are difficult to discern:

-	**Ambiguity in Examples**: It’s unclear what new insights the introductory examples provide. Established methods like CVaR-DRO (Example 3 in [1]) and chi-squared divergence DRO (Proposition 1 in [2]) already use duality forms, such as equations (1.14)–(1.17) and (1.2)–(1.5) being special examples. While the least squares and quantile regression examples appear novel, they lack clear interpretation. A discussion of how the robust model framework alters our perspective on these standard regressions and other cost functions would clarify the framework’s value (e.g., a new perspective?).

-	**New interpretations in Sec 6**: The interpretation in Section 4 results are unclear. Much of this material appears to be standard in DRO literature or from standard DRO duality, and the equivalence in equations (6.4)–(6.6) is not sufficiently justified. In terms of examples, Specifically, terms $R_{\phi, \beta}$ in (6.4), (6.7), (6.10) are not clearly explained. If these are defined based on Sec 3, should’t they follow directly from the Definition 3.1? Besides, I am struggling to find the connections between this and the risk quadrangle framework. If the intent is to show this framework is more general, then the authors should provide concrete examples illustrating this generality and explain why aspects like negative $Q$ values are important.
# General Comments
-	**Suitability for ICLR**: Given its current form, I am uncertain about this paper’s suitability for an ML-focused conference like ICLR. The risk quadrangle framework may be too theoretical for a general ML audience, and the connection to robust optimization is unclear in terms of practical ML relevance.
-	**Notations**: The paper’s notation can be streamlined. For example, similar terms like $Q_{\phi, \beta}^R$ (Line 39), $Q_{\phi, \beta}^V$ (line 104), $Q_{\phi,\beta}$ (Lie 288) represent similar concept. A unified notation would improve readability.

Reference:
[1] John Duchi, Hongseok Namkoong. Learning Models with Uniform Performance via DRO. Annals of Statistics. 2020.
[2] Henry Lam. Sensitivity to serial dependency of input processes: A robust approach. Management Science. 2018.

**Questions:**

See the weakness above and another clarification question:

-	Between Line 94 and 100, what is the choice of $\lambda$ here, it should be $\sqrt{\beta}$ right?

---

> ### Author Response · Authors · 2024-11-25
>
> Thank you for taking the time to carefully read through the unclear sections multiple times and sharing your valuable suggestions. We have made a serious effort to improve the clarity based on your comments.
>
> **Organization**
>
> - We have completely rewritten the Introduction to present the background, motivation, contributions, and literature review in a logical sequence.
>
>   The two examples have been removed. The revised version starts by connecting DRO to FRQ through coherent risk measures and discusses the natural idea of integrating DRO into FRQ. We then raise the issue of non-coherency in some risk measures, such as the important mean-standard-deviation risk measure, which motivates the introduction of the novel extended $\varphi$-divergence. The paper's contributions are now better summarized in a paragraph on the Main Contributions.
>
>  -  We have added comments and examples to the definitions and theorems in Sec 2 to clarify the intuition and implications.
>
>     Comments have been added to explain the intuition behind each axiom of the quadrangle elements, including a concrete example from the Mean Quadrangle to help readers grasp the concept. For example, $\mathbb{E}[X]+\lambda\sigma(X)$ is presented as an important example of a risk measure. We have also added comments to the regression theorem to explain how it connects regression with DRO.
>
>     The technical discussion from the former Sec 2.3 on functional spaces has been moved to the appendix.
>
>     Regarding the axioms of deviation, regret, and error (former Defs 2.2, 2.3, 2.4), the theorem on the dual representation of the extended $\varphi$-divergence quadrangle refers to and verifies that they are satisfied. All subsequent examples also satisfy these axioms. We have kept these axioms as preliminaries to maintain the completeness of the structure.
>
>  - We have reorganized the sections for a more coherent structure.
>
>     In the current version, Sec 3 contains the technical results.  Sec 4 contains concrete examples. Sec 5 contains the interpretation and concrete examples.
>
>     Sec 3, 4, and 5 now flow naturally: Sec 3.1 introduces the extended $\varphi$-divergence risk measure and completes the risk quadrangle for the defined risk measure in dual representation. Sec 3.2 derives the primal representation based on Sec 3.1. Sec 4 provides concrete examples of the extended $\varphi$-divergence quadrangle in primal representation. Sec 5.1 uses the dual representation from Sec 3.2 for a RO/DRO interpretation, and Sec 5.2 presents concrete examples of learning tasks, using the examples from Sec 4 and the interpretations from Sec 5.1.
>
>  - We have added non-technical comments after theorem statements to explain their implications.
>
>     For Theorem 3.1 (Extended $\varphi$-Divergence Quadrangle), we write: "After the discussion of the $\varphi$-divergence ambiguity set and the risk envelope $\mathcal{Q}$ in Section 3.2, it will be clear that Theorem 3.1 integrates DRO into the FRQ framework. The coherent risk measure in DRO is a special case of the extended $\varphi$-divergence risk measure. New quadrangles can be constructed by plugging extended $\varphi$-divergences into Definition 3.3. The dual representation provides a robust optimization interpretation for many well-known optimization problems (Section 5)."
>
>     For Theorem 3.2 (former 4.1) (Primal Extended $\varphi$-Divergence Quadrangle), we write:" The quadrangle elements in primal representation facilitates optimization, since the minimax prob- lem of minimizing the worst-case expectation becomes a minimization with additional scalar variable(s). Furthermore, substituting important extended $\varphi$-divergence functions into the definitions, we recover many risk quadrangles with interpretable expressions (Section 4)."
>
>     For Proposition 6.2 (former 7.2), we write:" Proposition 6.2 allows us to directly calculate the risk identifier (worst-case weight) given the solu- tion to the problem in primal representation. It will be used for calculation in Section 8."
>
>     For Proposition 7.1 (former 8,1), we write:" This study starts with developing new risk measures given a $\varphi$-divergence function. There exists a duality between divergence and risk that allows us to recover the $\varphi$-divergence from the elements of the corresponding $\varphi$-divergence quadrangle."
>
>     Propositions 6.1 and 7.1 are not used in this study. They provide insights into the conditions satisfied by the statistic and the duality between divergence and risk.

---

> ### Author Response · Authors · 2024-11-25
>
> **Contribution**
>
> We have updated the Main Contributions paragraph in the Introduction to provide a clearer summary. Since this issue was raised by all reviewers, we have also included it in the general comment section.
>
>   - The primal and dual representations of the objective functions in CVaR-DRO and $\chi^2$-DRO are concerned only with the $\varphi$-divergence risk measure, not the complete quadrangle. Thus, the interpretation of quantile regression and least squares regression as DRO/RO is not established in the referenced literature.
>
>     This study  provides a new perspective that the regression problems themselves can be viewed as DRO/RO, where the  random loss  is the residual. Current literature on further robustifying regression and classification may benefit from the insight that the original problems are already  DRO/RO.
>
>     Moreover, in the case of $\chi^2$-DRO, it is known that mean-standard deviation risk measure is an upper bound of the worst-case expectation under $\chi^2$-divergence.  In the updated Sec 3.4 and Example 2 and 6, we demonstrate the following: $(i)$ the mean-standard deviation risk measure is associated with the \textit{extended} $\chi^2$-divergence. $(ii)$ The result on upper bound is a special case of the relation between the extended $\varphi$-divergence and the extended version. $(iii)$ The upper bound becomes equality when $\beta$ is sufficiently small. $(iv)$ DRO with $\chi^2$-divergence ambiguity set, in fact, minimizes the second-order superquantile.
>
>
>   - We would like to clarify that it is not previously known that regression problems themselves are directly connected with DRO/RO, where the random loss is the residual without intercept, $Y-f(\tilde{X})$.
>
>      The equivalence of (former) 6.4 and (former) 6.5, 6.6 follows from using the dual representation (former) 3.1, and using the negative margin $-L(w,b)$ as the random loss $X$.  Indeed, $\mathcal{R}_{\varphi,\beta}$  in (former) 6.4, 6.7  directly follows from (former) 3.1. and  $\mathcal{E}$ in (former) 6.10 from (former) 3.4.
>
>      (Former) 6.4, 6.7, and 6.10 correspond to widely used learning tasks. The purpose of demonstrating equivalence is to show that these tasks can be viewed as RO/DRO through dual representation. We would like to emphasize a key connection to FRQ: the regression problem (error minimization) is not risk minimization and, therefore, cannot directly be interpreted as RO/DRO. The equivalence among (former) 6.10, 6.11, and 6.12 holds due to the regression decomposition theorem, which was not made sufficiently clear in the original manuscript. In the updated version, we clarified this by introducing Theorem 2.1 (Error Shaping Decomposition of Regression) and referencing it in Sec 5.
>
>     The aspect of negative $Q$ is important because it allows the risk quadrangle to encompass important examples. The most notable example is the mean-standard deviation risk measure from the Mean Quadrangle generated by the extended $\chi^2$-divergence. In the literature, this risk measure was only connected to RO/DRO through inequality or asymptotic relations [1,2,3].
>
>
> **General Comments**
>
>  - In the updated version, we provide intuitions and examples to better illustrate the theoretical framework. We show that common learning tasks in classification and regression can be viewed as RO/DRO with the extended $\varphi$-divergence ambiguity set, which brings a new perspective on the problems.
>
>  - In  3.6 of the updated version, we streamline the notation.  $\mathcal{Q}^1_{\varphi, \beta}$ and $\mathcal{Q}_{\varphi, \beta}$ differs by an additional condition $\mathbb{E}[Q]=1$,   which is reflected in the superscript.
>
>  - Yes. We removed $\lambda$ and replaced it by $\sqrt{\beta}$.
>
>
> We sincerely appreciate your detailed review. We hope that we have effectively addressed the concerns raised. We are happy to provide further information if needed.
>
>
> References:
>
> [1] Lam, H. (2016). Robust sensitivity analysis for stochastic systems. Mathematics of Operations Research, 41(4):1248–1275.
>
> [2] Duchi, J. and Namkoong, H. (2019). Variance-based regularization with convex objectives. Journal of Machine Learning Research, 20(68):1–55.
>
> [3] Kuhn, D., Shafiee, S., and Wiesemann, W. (2024). Distributionally robust optimization.

---

> > ### Comment · Reviewer_Bojo · 2024-11-25
> >
> > Thank you for your detailed feedback and consideration in conducting a major paper revision. Similar to the opinion of Reviewer aJtz, I feel like the main body contains many new things, e.g. examples in Sec 4. Therefore, I believe that the paper may require another round of full reviews by polishing it further.

---

### Author Response · Authors · 2024-11-25

We sincerely thank all reviewers for their thoughtful and detailed feedback. Your comments have been highly constructive and have greatly helped us improve the quality of our paper.

The primary concerns raised were related to readability, motivation, and clarity of contributions. We have made significant efforts to address these concerns comprehensively.

- **Readability.** We have undertaken a major revision of the paper to improve its coherence. Each section now builds logically on the previous ones:
  - Sec 1 Introduction presents the background, motivation, contributions, and literature review in a reader-friendly order.
  - Sec 2 introduces the necessary background on $\varphi$-divergence risk measure and FRQ.
  - Sec 3.1 introduces the extended $\varphi$-divergence risk measure, then completes the risk quadrangle.
  - Sec 3.2 and 3.3 explore the relation between the $\varphi$-divergence quadrangle and its extended version, and their connection to RO/DRO.
  - Sec 3.4 derives the primal representation of the extended $\varphi$-divergence quadrangle from the dual representation in Sec 3.1.
  - Sec 4 derives important examples of extended $\varphi$-divergence quadrangles from the primal representation.
  - Sec 5 provides the RO/DRO interpretation for various learning tasks using the dual representation in Sec 3.1, and provides two important examples.
  - Sec 6 provides a way to compute the worst-case weight using the optimal solution from the primal representation.
  - Sec 8 visualizes the worst-case weight in various tasks for the Mean Quadrangle in Sec 5, using the calculation method in Sec 6.

- **Motivation.**
  We have added comments throughout the paper explaining the intuition behind definitions and the implications of theorems:
  - We introduce the motivation of our study in the Introduction. The Introduction starts with connecting DRO to FRQ through coherent risk measure, and discusses the natural idea of integrating DRO into FRQ. We then raise the issue of non-coherency of some important risk measures, such as the mean-standard-deviation risk measure, which motivates the introduction of the novel extended $\varphi$-divergence.
  - We explain the intuition behind the axioms of risk quadrangle elements, and exemplify the elements with the important Mean Quadrangle. We also comment on the implication of regression theorem on connecting regression with DRO.
  - We added comments to definitions and theorems to explain the implications. For example, for Def 2.8 and 3.1, and Theorem 3.1 and 3.2.

- **Contribution.**
  We rewrite the paragraph Main Contributions. Our main contributions are as follows:
  - **Extension of $\varphi$-divergence:** We define the extended $\varphi$-divergence and its associated risk measure, allowing for negative values in the worst-case weight. The extension recovers risk measures commonly used as objective functions across various tasks. A notable example is the mean-standard deviation risk measure associated with the extended $\chi^2$-divergence.
  - **Completion of Quadrangle:** For the extended $\varphi$-divergence risk measure, we complete the risk quadrangle and derive primal and dual representations for risk, deviation, regret, and error. The primal representation facilitates convex optimization formulations. The dual representation provides a robust optimization (RO) interpretation for measures associated with the extended $\varphi$-divergence, and a DRO interpretation for those associated with the $\varphi$-divergence. The RO objective functions are upper bounds (conservative version) for their DRO counterparts. A well-known special case is that the mean-standard deviation risk measure bounds the $\chi^2$-divergence risk measure.
  - **Examples and Interpretation:** We provide a range of examples to illustrate that the extended $\varphi$-divergence quadrangle recovers many important quadrangles. The quadrangle elements are used as objective functions in various learning tasks, such as least-squares regression, quantile regression, support vector machines, and CVaR optimization. Through the dual representation, these tasks have a novel interpretation as robust optimization.

We sincerely thank all reviewers for their constructive feedback. We hope that these major revisions address the concerns effectively. We welcome any additional suggestions for further improving the paper.

---

### Author Response · Authors · 2024-11-29

With the extended discussion period, we would greatly appreciate any additional feedback on areas where the paper could be further improved. We kindly request reviewers to consider adjusting the rating to reflect the contributions and addressed concerns, or to update the confidence if the revisions have not been fully reviewed. Thank you very much for your time!

---

### Meta-Review · Area_Chair_Q5xy · 2024-12-09

**Metareview:**

This paper extends the Fundamental Risk Quadrangle (FRQ) framework by integrating distributionally robust optimization (DRO) based on an extended $\varphi$-divergence. The authors derive primal and dual representations of the quadrangle elements (risk, deviation, regret, error, and statistic), offering new interpretations for problems like regression, classification, and portfolio optimization as robust optimization tasks.

While reviewers appreciate the theoretical contributions, they express concerns about the paper's clarity, structure, and the extent of the revisions made. The paper was considered overly technical in parts, with unclear explanations and a confusing organization that hindered understanding of the core contributions.

**Additional Comments On Reviewer Discussion:**

Although the authors revised the paper to improve readability and provide more commentary on key concepts, reviewers felt the changes were too extensive and that the paper requires a full review cycle to properly assess the revisions.

---

### Decision · Program_Chairs · 2025-01-22

Reject